



# A new approach for measuring the UV-Vis optical properties of ambient aerosols

Nir Bluvshtein, J. Michel Flores, Lior Segev, Yinon Rudich

Department of Earth and Planetary Sciences, Weizmann Institute of Science, Rehovot 76100, Israel

5  *Correspondence to:* Yinon Rudich (yinon.rudich@weizmann.ac.il)

**Abstract.** Atmospheric aerosols play an important part in the Earth's energy budget by scattering and absorbing incoming solar and outgoing terrestrial radiation. To quantify the effective radiative forcing due to aerosol–radiation interactions, researchers must obtain a detailed understanding of the spectrally dependent intensive and extensive optical properties of different aerosol types. Our new approach obtains the optical coefficients and the single scattering albedo of the total aerosol population over 300-650 nm wavelength, using a broadband cavity-enhanced spectrometer (extinction), a photoacoustic cell coupled to a cavity ring down spectrometer (extinction and absorption), and a nephelometer (scattering). Combining these coefficients with aerosol size distribution data, we retrieved the time- and wavelength-dependent effective complex refractive index of the aerosols. Retrieval simulations and laboratory measurements of brown carbon proxies showed low absolute errors and good agreement with expected and reported values. Finally, we utilized our new broadband method to achieve continuous spectral and time-dependent monitoring of an ambient polydisperse aerosols population, including, for the first time, extinction measurements using cavity enhanced spectrometry in the 315 to 345 nm UV range, in which significant light absorption may occur.

Keywords: Atmospheric aerosol, optical properties, spectral dependence, refractive index, absorption, scattering, cavity enhanced spectroscopy, brown carbon.

## 1 Introduction

Atmospheric aerosols affect the Earth's energy balance directly via their interaction with incoming solar radiation and indirectly by altering cloud microphysical and optical properties. The effective radiative forcing (EFR) due to aerosol–radiation interactions (ERFari) encompasses attenuation of solar flux to the surface due to direct scattering and absorption and rapid adjustments of the atmospheric temperature profile (IPCC, 2013). The latter is due to energy released as heat by light-absorbing aerosols that can influence cloud lifetime (Hill and Dobbie, 2008; Davidi et al., 2009; Allen and Sherwood, 2010; Koch and Del Genio, 2010; Nabat et al., 2014). Irradiation changes from ERFari and the EFR due to aerosol–cloud interactions (ERFaci) are still two of the largest uncertainties in our understanding of anthropogenic radiative forcing (IPCC, 2013).





Radiative transfer models use aerosol optical depth, single scattering albedo (*SSA*), the scattering phase function, and the asymmetry parameter to describe the interaction between aerosols and solar radiation. An accurate representation of the complex refractive index (RI; $m = n + ik$), which is an intensive optical property of aerosol types and components, is required to calculate these parameters.

Aerosol optical properties are typically measured as scattering, absorption, or extinction coefficients ($\alpha_{sca}$, $\alpha_{abs}$ and $\alpha_{ext}$, respectively) by a variety of *in situ* techniques. These include integrating and reciprocal nephelometry (Nakayama et al., 2010), filter-based absorption measurements (Guyon et al., 2003; Zhang et al., 2013), photoacoustic spectrometry (PAS) (Lack et al., 2012), extinction cells, and cavity enhanced spectrometry (CES) (Varma et al., 2013). Many of these methods are restricted to a single or a few discrete wavelengths and their ability to provide wavelength-dependent measurements is

limited. Broadband CES instruments were recently developed for aerosol extinction measurements at wavelength ranges of about 30 to 40 nm per cavity (Washenfelder et al., 2013; Zhao et al., 2013; Flores et al., 2014; Washenfelder et al., 2015). White-type extinction cells with a UV-Vis light source and a grating spectrometer were recently used for aerosol extinction measurements over a wide range of wavelengths from below 250 nm up to 700 nm. However, white-type extinction cells suffer from low optical path length (tens of meters, as opposed to several kilometers in CES instruments) which leads to the

detection limit being an order of magnitude lower than that of CES (Chartier and Greenslade, 2012; Jordan et al., 2015). Organic particulate matter that have strong wavelength-dependent light absorption characteristics, with higher absorption at near-ultraviolet and blue wavelengths (Andreae and Gelencser, 2006; Laskin et al., 2015), are known as atmospheric brown carbon (BrC). BrC is mostly composed of anthropogenic or biogenic secondary organic aerosols and aerosols from biomass burning (Spracklen et al., 2011). The contribution of BrC to radiative forcing still poses one of the largest uncertainties in

our understanding of climate forcing. BrC may be the dominant light absorber downwind of industrial areas and in biomass burning plumes (Feng et al., 2013). In the atmosphere, it is found internally or externally mixed with inorganic particles and black carbon (Cappa et al., 2012). If internally mixed with black carbon, BrC may cause absorption enhancement through the lensing effect (Bond et al., 2006). A better quantification of the spectral dependency of optical properties of BrC aerosols is required in order to reduce the uncertainty surrounding the ERFari.

In this study, we present a new approach for retrieving a broadband UV-Vis spectrum (300 to 650 nm) of the total aerosol population, including its $\alpha_{sca}$, $\alpha_{abs}$ and $\alpha_{ext}$, its *SSA*, and, together with size distribution measurements, its effective complex RI. The retrieval method utilizes a combination of several different instruments: CES, PAS, and nephelometer. We validate the method with computer simulations and laboratory measurements and show how the method can be applied for continuous spectral and time-dependent monitoring of an ambient polydisperse aerosols population. To the best of our

knowledge, we report the first implementation of broadband CES for aerosol extinction measurements in the 315 to 345 nm UV range (being a range in which significant light absorption may occur) for laboratory and ambient aerosols.



## 2 Methods

### 2.1 Approach

To obtain the optical properties of light-absorbing ambient aerosols in the 300 to 650 nm wavelength range, we used measurements of $\alpha_{ext}$, $\alpha_{sca}$ and $\alpha_{abs}$. The extinction coefficients were obtained with a homebuilt broadband cavity-enhanced

spectrometer (BBCES) measuring at two distinct wavelength ranges: 315 to 345 nm and 360 to 390 nm. Extinction and absorption coefficients at $\lambda = 404$ nm were measured using a homebuilt multi-pass PAS (Lack et al., 2012) coupled to a cavity ring down (CRD) spectrometer (Bluvshtein et al., 2012; Flores et al., 2012b) (PA-CRD-S). The scattering coefficients at 457, 525, and 637 nm were measured using an integrating nephelometer (IN100, AirPhoton, USA). The $\alpha_{ext}$, $\alpha_{sca}$, and $\alpha_{abs}$, measurements were used together with the aerosol size distribution and the aerosol number concentration in a novel

procedure to obtain the broadband optical coefficients, the SSA, and the broadband effective complex RI in the 300 to 650 nm wavelength range. The procedure is described in detail in Sect. 2.3. The term effective complex RI is used to represent the whole particle size distribution. It is the complex RI from which, for the corresponding size distribution, we derive the optical coefficients that agree most closely with the measured or input values.

### 2.2 Instrumentation

#### 2.2.1 Broadband cavity-enhanced spectroscopy

We use a dual channel BBCES to measure the aerosol optical extinction between 315 to 345 nm and 390 to 420 nm (at a 0.5 nm resolution). The 315 to 345 nm cavity uses a new, laser driven Xe lamp, and its design is similar to that described recently by Washenfelder et al., (2016). The 390 to 420 nm cavity is as described in Flores et al., (2014). Only a brief description and the main differences are highlighted here.

For the 390 to 420 nm cavity (BBCES-407), we use a light emitting diode (LED) centered at 407.1 nm with a measured optical power output of 0.450 W (M405D2, Thorlabs, Newton, NJ, USA). The LED is temperature-controlled and powered by a constant-current power supply to achieve a stable optical power output. The output from the LED is collimated using a single F/1.2 fused silica lens and optically filtered using a bandpass filter (FB400-40, Thorlabs, Newton, NJ, USA) before entering the optical cavity formed by two mirrors, 2.54 cm in diameter and 1 m radius of curvature (Advanced Thin Films,

Boulder, CO, USA). For the 315 to 345 nm cavity (BBCES-330; see Washenfelder et al., (2016) for a detail description), we use a broadband light source (EQ-99FC LDLS; Energetiq, Woburn, MA, USA) consisting of a continuous wave diode laser at 974 nm that pumps a Xe plasma (Islam et al., 2013). The light source is air-cooled and temperature-controlled using water circulation through an attached aluminum plate to prevent intensity drifts. This light exits through a 600 μm optical fiber and it is collimated and coupled into the optical cavity using an off-axis parabola with a 0.36 numerical aperture (RC04SMA-

F01; Thorlabs, Newton, NJ, USA). To remove stray light, the light passes through two colored glass filters (Schott Glass WG320 and UG11) before entering the cavity. This cavity also consists of two mirrors, 2.54 cm in diameter and 1 m radius of curvature (Layertec GmbH, Mellingen, Germany).





The typical measured mirror reflectivity for the BBCES-330 and BBCES-407 cavities is 0.99960 and 0.99994 at 330 and 420 nm, respectively. After exiting each cavity, the light is directly collected using a 0.1 cm F/2 fiber collimator (74-UV, Ocean Optics, Dunedin, FL, USA) into one lead of a two-way 100 μm core HOH-UV-VIS fiber (SR-OPT-8015, Andor Techonology, Belfast, UK) that is linearly aligned along the input slit of the grating spectrometer.

The spectra are acquired using a 163 mm focal length Czerny-Turner spectrometer (Shamrock SR-163, Andor Technology, Belfast, UK) with a charge coupled device (CCD) detector (DU920P-BU, Andor Technology, Belfast, UK) maintained at -50 °C. The spectrometer is temperature-controlled at 32.0 ± 0.1 °C. Dark spectra are acquired with the input shutter (SR1-SHT-9003, Andor Technology, Belfast, UK) closed prior to each set of spectra. The wavelength is calibrated using a Hg/Ar pen-ray lamp.

The $\alpha_{ext}$ of the aerosol is determined as the difference in light intensity between a filled cavity and a particle-free cavity, taking into account the mirror reflectivity and the Rayleigh scattering of the carrier gas (Washenfelder et al., 2013).

### 2.2.2 Photoacoustic spectrometer coupled to a cavity ring down spectrometer

A single wavelength PA-CRD-S (Fig. 1) is used to directly measure both $\alpha_{ext}$ and $\alpha_{abs}$ at $\lambda = 404$ nm. The PA-CRD-S system described in this section (Fig. 1) is composed of a 110 mW 404 nm diode laser (iPulse, Toptica Photonics, Munich,
Germany) modulated at the measured PAS resonance frequency on a 50% duty cycle. The laser beam (blue arrow in Fig. 1) is split into two separate optical paths (directed to the CRD-S and the PAS, respectively) using a variable beam splitter composed of a quarter waveplate (¼λ) and a polarizing beam splitter (PBS). With the current setup, turning the ¼λ between $0^0$ and $90^0$ varies the intensity ratio between the two optical paths from 0:1 to 1:1 CRD-S to PAS, respectively. The beam directed to the PAS is turned and aligned into the PAS cell through a set of two plano-convex lenses (focal lengths of 30 mm
and 50 mm) used as a telescope in order to collimate the beam into a diameter of about 1.5 mm. The beam directed to the CRD-S passes through another ¼λ plate, which together with the PBS serves as a variable attenuator protecting the laser head from the beam reflected back by the highly reflective mirror of the CRD-S. This back-reflected beam is transmitted through the PBS into a photodiode (PD) (as shown by the dashed arrow in Fig. 1), with the PD serving as an external trigger source for the CRD-S decay measurement. The forward beam is then turned and aligned into the CRD-S cavity by a set of
turning mirrors. While the sensitivity of the PAS is related to the power intensity of the laser, the CRD-S system requires only the minimal laser power needed by the PD. This allows us to divert approximately 78% of the laser power (about 86 mW) to the PAS cell and thus optimize its sensitivity.

### 2.2.2a Photo acoustic spectrometer

In a PAS, modulated laser light is absorbed by a sample of particles or gas, generating a modulated acoustic wave whose
intensity is proportional to the energy absorbed by the sample. This acoustic wave, which is detected by a sensitive microphone, has a characteristic radial and longitudinal resonance when the light source is modulated at the cavity resonance



frequency. For a more detailed description of the PAS method for aerosol light absorption measurement see Arnott et al., (1999) and Nagele and Sigrist, (2000).

We use a multi-pass astigmatic PAS cell (see Lack et al., (2012) for a detailed description). Briefly, the PAS is composed of dual half-wavelength resonators (11 cm long, 1.9 cm diameter) capped on either end with 1/4 wavelength acoustic notches.

The total sample cell volume is 185 cm$^3$. While both resonators are open to sample flow, only one is exposed to the modulated laser light; the other is used for noise cancellation. Microphones are placed at the antinode of the sound wave in the center of each resonator and the speaker is placed at the background resonator. The resonance frequency specific for this system is found by producing white noise using a speaker in the reference resonator. Each segment is sampled by the microphones at a 100 kHz rate and analyzed by a fast Fourier transform algorithm.

The astigmatic optical configuration consists of two 5.08 cm diameter, high reflectivity mirrors (ARW Optical, Wilmington, NC, USA; dielectric coating R > 99.5%) spaced 35 cm apart and mounted on adjustable mirror mounts. The laser side mirror has a cylindrical radius of curvature of 43 cm and a 2 mm hole drilled in the center. The back mirror has a cylindrical radius of curvature of 47 cm, and is rotated 90° to the radius of curvature of the laser side mirror. Astigmatic alignment is achieved by aligning the laser through the 2 mm hole drilled in the center of the first mirror and onto an off-center target on the second

mirror. Each following reflection should also be directed to an off-center target on the other mirror. The PAS cell is mounted within the path of the laser multi-pass. The laser light passes through the PAS cell via two 1 mm thick windows (CVI Laser, Albuquerque, NM, USA), each with a high transmissivity (T > 99.5%) antireflective coating. The laser power is continuously monitored and used to cancel variations in acoustic signal related to laser power fluctuations.

The PAS calibration procedure is described in detail elsewhere (Bluvshtein et al., in preparation). In short, the complex RI of

dry nigrosine films is measured at 404 nm using spectroscopic ellipsometry and transmission (Hilfiker et al., 2008). Nigrosine dye is dissolved into an aqueous solution and nebulized using a constant output atomizer (model 3076, TSI, 35 psi, flow of 2.5 standard liters per minute (SLM)), with dry particle-free nitrogen, generating a polydispersed distribution of droplets. The aerosol population is subsequently dried (relative humidity (RH) < 5%) using two silica gel diffusion dryers and size-selected with an electrostatic classifier (differential mobility analyzer (DMA) model 3085, TSI), operating with a

particle-free, dry nitrogen sheath flow of 3 to 15 SLM. A 10:1 ratio of sheath flow to sample flow is maintained. An impactor is used on the DMA inlet to reduce the contribution from multiply charged particles. Nigrosine particles of several sizes (200 to 400 nm mobility diameter) and number concentrations (counted by a condensation particle counter; CPC; model 3775, TSI) were flown through the PAS cell and its signal was compared to the aerosol $\alpha_{abs}$ calculated using the complex RI retrieved from the dry film measurements and a Mie algorithm.

**2.2.2b Cavity ring down spectrometer**

The CRD-S method has been extensively described in previous publications (Sappey et al., 1998; Vander Wal and Ticich, 1999; Smith and Atkinson, 2001; Bulatov et al., 2002; Thompson et al., 2002; Strawa et al., 2003; Pettersson et al., 2004; Riziq et al., 2007; Bluvshtein et al., 2012). Briefly, a single wavelength laser light source is modulated and coupled into the





high-finesse optical cavity. The cavity transmission is coupled in to an optical fiber and focused onto a photomultiplier tube (PMT), which measures the decay of the light intensity due to aerosol absorption and scattering (extinction). Measuring the light time constant, with an empty cavity and with a cavity filled with aerosols, allows the direct measurement of $\alpha_{ext}$.

The two optical cavities of the BBCES and the optical cavity of the CRD-S were assembled together in a rigid optical cage

to minimize alignment and stability issues. Aerosol flow was introduced to the center of each cavity and pulled from its sides (Fig. 1, downward arrows into the CRD and BBCES). Using this setup eliminated a significant source of error in determining the extinction coefficient, namely, uncertainty in the length of the aerosol sample within the cavity length (Miles et al., 2011). The optical cage and the PAS cell were constructed on vibration-isolated breadboard in a temperature controlled environment (23 ± 0.25 °C).

### 2.2.3 Integrating Nephelometer

The AirPhoton IN100 integrating nephelometer (IN) is a component of the global Surface PARTiculate Aerosol Network (SPARTAN), whose purpose is to evaluate and enhance satellite based estimates of ground level particulate matter. The nephelometer is a continuous sampling, LED-based device measuring total $\alpha_{sca}$ at three optical channels (red, green and blue) centered on 637 nm, 525 nm, and 457 nm over an angular range of 7 to 170° (Snider et al., 2015).

### 2.2.4 Size distribution and number concentration

Aerosol number concentration is measured with a condensation particle counter (CPC; Model 3775, TSI) and the particle size distributions are obtained by a scanning mobility particle sizer (SMPS; 3085 DMA and 3775 CPC, TSI).

### 2.3 Retrieval methodology

We have developed a two-step method to derive continuous values of $\alpha_{ext}$, $\alpha_{sca}$, $\alpha_{abs}$ and *SSA* in the 300 to 650 nm wavelength

range. Using these values of $\alpha_{ext}$, $\alpha_{sca}$, and $\alpha_{abs}$ together with the measured aerosols size distribution and the number concentration, we also retrieve the effective complex RI. A flow chart of the retrieval methodology is shown in Fig. 2.

### 2.3.1 Broadband extinction, scattering, absorption, and SSA retrieval methodology

First, the extinction data from the BBCES measurements (315 to 345 nm, 390 to 420 nm) are fitted with a power law function. This fit is used to extrapolate $\alpha_{ext}$ to the nephelometer wavelengths. Next, data for $\alpha_{abs}$ at 404 nm are used together

with an initial guess of a power law coefficient to calculate $\alpha_{abs}$ at the three nephelometer wavelengths (637 nm, 525 nm, and 457 nm). These three $\alpha_{abs}$ values, together with the three $\alpha_{ext}$ values from the power law fit of the extinction data, are used to calculate $\alpha_{sca\_calc}$ at the nephelometer wavelengths. Then the minimum square difference ($\chi^2$) is calculated between $\alpha_{sca\_calc}$ and the measured $\alpha_{sca\_meas}$; the power law coefficient used to extrapolate the $\alpha_{abs}$ data is varied iteratively until the minimum difference between $\alpha_{sca\_calc}$ and $\alpha_{sca\_meas}$ is found (Fig. 2a). This procedure is repeated with an exponential function to

extrapolate $\alpha_{ext}$ and with an exponential coefficient to extrapolate $\alpha_{abs}$ and two additional times to cover all four possible





combinations of exponential and power law representations of $\alpha_{ext}$ and $\alpha_{abs}$. At each repetition, the minimum difference between $\alpha_{sca\_calc}$ and $\alpha_{sca\_meas}$ is found iteratively and the global minimum difference from the four combinations is selected. This information is used to calculate $\alpha_{ext}$ and $\alpha_{abs}$ in the wavelength range of 300 to 650 nm (Fig. 2b), and $\alpha_{sca}$ is then calculated by subtraction. Finally, the size-weighted $SSA$ is calculated as: $SSA = \alpha_{sca} / \alpha_{ext}$.

### 2.3.2 Methodology for retrieving the effective complex RI of the total particle size distribution

Using the retrieved $\alpha_{ext}$ and $\alpha_{sca}$ described in Sect. 2.3.1 together with the measured size distribution and the aerosol number concentration, the effective complex RI of the total particle size distribution is retrieved at each individual wavelength. The effective complex RI in the context of this work is one that, for a given size distribution and number concentration, would satisfy a minimum difference between the theoretical values of $\alpha_{ext}$ and $\alpha_{sca}$ (based on a Mie theory calculation) and the extrapolated or measured $\alpha_{ext}$ and $\alpha_{sca}$. A specialized Mie theory algorithm was written in order to retrieve the effective complex RI for the total size distribution of the particles. Briefly, an array of initial guesses is used to initiate an iterative converging search of a theoretical complex RI for which a Mie calculation (Bohren and Huffman, 1983) with a given size distribution, number concentration, and wavelength produces a best fitted pair of theoretical $\alpha_{ext}$ and $\alpha_{sca}$ (see Fig. 2b). The best fit is determined by minimizing the $\chi^2$ function:

$$\chi^2 = \frac{\left(\frac{\alpha_{ext\_meas}}{N} - \sigma_{ext\_calc}\right)^2}{\left(\frac{d\alpha_{ext}}{N}\right)^2 + \left(\frac{-\alpha_{ext} \times dN}{N^2}\right)^2} + \frac{\left(\frac{\alpha_{sca\_meas}}{N} - \sigma_{sca\_calc}\right)^2}{\left(\frac{d\alpha_{sca}}{N}\right)^2 + \left(\frac{-\alpha_{sca} \times dN}{N^2}\right)^2} \tag{1}$$

were $\alpha_{ext\_meas}$ and $\alpha_{sca\_meas}$ are the retrieved or measured $\alpha_{ext}$ and $\alpha_{sca}$, $N$ is the particle number concentration , $\sigma_{ext\_calc}$ and $\sigma_{sca\_calc}$ are the theoretical extinction and scattering cross sections weighted by the size distribution, and $d$ denotes the uncertainty on the associated parameter.

To estimate the retrieval uncertainties in $n$ and $k$ ($\Delta n$ and $\Delta k$), the algorithm returns the values of $n$ and $k$ that satisfy $\chi_0^2 \leq \chi^2 \leq \chi_0^2 + 1$ where the value 1 denotes 1σ deviation from the minimum $\chi^2$ ($\chi_0^2$) (in the case of one degree of freedom) (Press et al., 1992).

### 2.4 Validation of the retrieval methodology

In order to test this new approach and evaluate its merits, we performed three different tests: 1) a computer simulation of time dependent extinction, scattering and absorption measurements at the instruments' wavelengths; 2) laboratory measurements of two BrC proxy materials; and 3) a 24 hour ambient aerosol measurement.

### 2.4.1 Computer simulation

For the computer simulation, 100 different synthetic data sets of complex RIs in the 300 to 650 nm range were composed at a resolution of 1 nm. The real part ($n(\lambda)$) ranged from 1.692 at 300 nm to 1.856 at 650 nm, and the imaginary part ($k(\lambda)$) ranged from $8.156 \times 10^{-2}$ at 300 nm to $1.781 \times 10^{-3}$ at 650 nm. Both $n(\lambda)$ and $k(\lambda)$ were scaled by two incoherent sine waves



to simulate temporal variability. A log-normal size distribution with a mode at 80 nm, a geometric standard deviation of 1.33, and a number concentration of $10^4$ cm$^{-3}$ were assumed to calculate $\alpha_{ext}$, $\alpha_{sca}$ and $\alpha_{abs}$ at the instrumental wavelengths. An additional error was assigned randomly out of a normal distribution with ±2% (1 standard deviation). The $\alpha_{ext}$, $\alpha_{sca}$, $\alpha_{abs}$, *SSA* and effective complex RI were then retrieved as described above for all the synthetic instrumental data sets. Figure 3 shows a

single modeled example of the procedure. The retrieved $\alpha_{ext}$, $\alpha_{sca}$, $\alpha_{abs}$, and *SSA* are shown in Fig. 3a and the broadband complex refractive index in Fig. 3b.

### 2.4.2 Laboratory measurements

For the laboratory measurements, two BrC proxy materials were used to test the retrieval method: Suwannee River Fulvic Acid (SRFA; 1S101F, International Humic Substances Society (IHSS), Saint Paul, MN, USA) and Pahokee Peat Fulvic Acid

(PPFA; IHSS). SRFA and PPFA (aquatic/terrestrial based fulvic acids standards) have been used as proxy materials for atmospheric BrC (Dinar et al., 2008) for several decades since macromolecular organic substances in aerosols began to be analyzed (Hoffer et al., 2006; Gaffney et al., 2015) and are recognized as similar to humic and fulvic acids. Although Graber and Rudich, (2006) highlighted significant chemical and physical differences between atmospheric humic-like substances and terrestrial or aquatic humic substances, for the purpose of simulating UV-Vis light absorption by atmospheric BrC,

terrestrial/aquatic based humic substances standards remain useful.

Each substance was separately dissolved into an aqueous solution and nebulized as described in Sect. 2.2.2a. A complementary N$_2$ flow of 1.3 SLM was added and mixed with the sample flow, which was then introduced to the IN and subsequently split equally to the SMPS, CPC, PAS, and a three-cavity optical cage that contained the CRD-S and BBCES (see Fig. 4). There was no additional dilution of the sample flow once it was introduced to the IN to avoid differences in the

sampled particle concentration between the different instruments. Aerosol temperature, pressure, and RH were measured continuously.

### 2.4.3 Ambient aerosol measurement

To demonstrate the application of the retrieval methods to field applications, ambient aerosols were sampled during a 24 hour period. Using conductive tubing, ambient air was pulled from the roof of the Department of Earth and Planetary

Sciences building at the Weizmann Institute of Science through a PM$_{10}$ sampling inlet. Sampled air was dried (RH < 17%) with diffusion dryers and pulled isokineticly through the PA-CRD-S and BBCES. A total flow of 16.7 SLM was pulled through the PM$_{10}$ inlet as specified by the manufacturer. The CRD-S and the two BBCES systems sampled at a 0.2 SLM flow rate and the PAS at a 0.6 SLM flow rate (Fig. 4). The flow scheme was set up in a manner that ensured isokinetic splitting at every tube junction. Sampling was undertaken continuously for a 24 hour period from 18:30 on June 21, 2015 to

18:30 of the following day (local time). While each instrument has a different sampling rate, in order to simplify data analysis, the data outputs of the PAS-CRD-S, BBCES, IN and CPC were set to represent 2 minute averages.



Because the IN continually samples untreated ambient aerosols as part of the SPARTAN network, it remained on the roof. For this reason, IN measurements had to be corrected for particle hygroscopic growth. Snider et al., (2015) suggest correcting for increased scattering due to the growth of humidified particles using the relative change in particle volume, but this does not take into account the decrease in effective RI of the particles due to water uptake. See the supplementary material for a detail description of the correction we used.

## 3 Results and discussion

### 3.1 Computer simulations

The results of the computer simulations are summarized in Fig. 5. Box plots of the absolute value of the percent errors for the retrieved variables from the 100 different synthetic data sets are shown. Overall, the retrieved values are in very good agreement with the synthetic data. Results show that expected errors in the size weighted $SSA(\lambda,t)$, $\alpha_{ext}(\lambda,t)$ and $\alpha_{sca}(\lambda,t)$ are less than 10% for the full spectral range and less than 5% in the 400 to 500 nm range. Expected errors in the real part of the RI are less than 1% throughout the entire spectrum. Relative errors in the imaginary part of the RI and in $\alpha_{abs}(\lambda,t)$ are less than 60% for the 300 to 400 nm range and are expected to grow with increasing wavelength as these parameters go to zero. Under the conditions of this simulation at 400 nm wavelength (namely; the complex RI, particle size distribution, and number concentration), a relative error of 60% translates into absolute errors of 1 to 3 Mm$^{-1}$ on $\alpha_{abs}$ and of 0.01 to 0.015 on $k$, respectively (see Fig. 3).

### 3.2 Laboratory measurements

Figure 6 shows the measured optical coefficients of PPFA particles as well as the retrieved products of the broadband coefficients (Fig. 6a). The retrieved complex RI is also shown (Fig. 6b). The retrieved values for RI obtained from size-selected aerosols (Lack et al., 2006; Riziq et al., 2007; Trainic et al., 2011; Bluvshtein et al., 2012; Flores et al., 2012a; Lavi et al., 2013; Washenfelder et al., 2013; Flores et al., 2014) using the BBCES-315 and CRD-S are overlaid on Fig. 6b. The imaginary part of the complex RI calculated from UV-Vis absorption measurements of the diluted aqueous solution (Sun et al., 2007) is shown as a shaded area. The upper and lower limits of this area represent the range of assumed material density (1.1 to 1.3 g cm$^{-3}$; IHSS, personal communication) used in this calculation. There is very good agreement between all the retrieved values, with a slight difference between them only with respect to the real part of the RI in the 315 to 345 nm range.

Similarly to the PPFA measurements, Fig. 7 shows the measured and retrieved valued for $\alpha_{ext}$, $\alpha_{sca}$, $\alpha_{abs}$, and $SSA$ obtained using SRFA as the BrC proxy material. Overlaid on Fig. 7b are the published complex RI values retrieved from laboratory-generated SRFA. These retrievals were obtained from extinction measurements on size-selected particles. The accuracy of the new effective RI retrieval process is improved by incorporating direct absorption (Zarzana et al., 2014) and scattering measurements.



### 3.3 Ambient aerosol measurement

We measured ambient aerosol during a 24 h period to demonstrate the usefulness of the new retrieval methods. To check the reliability of the retrieval method, we first compared the *SSA* from the direct $\alpha_{ext}$ and $\alpha_{abs}$ measurements (using the PA-CRD-S) with the retrieved *SSA* at 404 nm (calculated from retrieved $\alpha_{ext}$ and $\alpha_{sca}$, Fig. 8) taking into account the uncertainties in both variables. Figure 8 shows an excellent correlation (slope = 1.010 ± 0.075; $R^2$ = 0.913) between the measured and retrieved *SSA*. The good agreement between measured and retrieved *SSA* is an indication that the broadband extinction retrieval procedure has little to no error at this wavelength. The full spectral retrievals of $\alpha_{ext}(\lambda,t)$, $\alpha_{sca}(\lambda,t)$, $\alpha_{abs}(\lambda,t)$ and $SSA(\lambda,t)$ over time are shown in Fig. 9 and the retrieved effective complex RI is shown in Fig. 10. Discontinuity in the sampling data is due to routinely performed reflectivity and zero air measurements required to avoid data drifts. Figure 11 shows the evolution of the total number concentration and the SMPS size distributions, normalized to the concentration of the mode diameter.

Between 06:30 and 08:30 during the morning rush hour, a sharp increase in $\alpha_{abs}$ and in the size weighted *SSA* is not apparent in the extinction and scattering coefficients (Fig. 9). This could be related to increased emissions of ultrafine light-absorbing combustion particles from traffic. This is supported by the decrease in the aerosol size shown in Fig. 11 and by the fact that *SSA* is inversely related to particle size. The notable increase in the extinction and scattering coefficients seen between 10:00 and 11:00 (Fig. 9) corresponds to the increase in particle concentration seen in Fig. 11 and is probably due to relatively large particles, as the *SSA* values are relatively high.

Considerable variations in particle number concentration and size distribution occur during the daytime (Fig. 11), and are probably due to transportation on nearby roads. The SMPS size scan duration was set to 5 minutes while large concentration variations occurred at shorter time scales. Unlike the $\alpha_{ext}(\lambda,t)$, $\alpha_{sca}(\lambda,t)$, $\alpha_{abs}(\lambda,t)$ and $SSA(\lambda,t)$ data, the retrieval of the effective complex RI is strongly dependent on accurate representation of the size distribution and aerosol particle number concentration. For this reason, Fig. 10 shows effective RI retrieval results between midnight and 07:00, when variations in number concentration and size distribution data were not as frequent.

In a recent review, Moise et al., (2015) compiled the optical properties data of secondary organic aerosols (SOA) and BrC from laboratory and chamber experiments and from ambient measurements reported in various studies. The compiled data were reported based on formation, aging pathways, and chemical composition. Various measurements of laboratory- and chamber-generated anthropogenic/biogenic SOA at 405 nm wavelength produced complex RIs with $1.45 \leq n \leq 1.7$ and $0 \leq k \leq 0.04$. The retrieval presented in Fig. 10 mostly falls within the lower limit of this wide range. As Moise et al., (2015) pointed out, laboratory- and chamber-generated SOA are mostly not as oxidized as reported ambient aerosols and, although some studies showed a possible inverse relation between the real part of the complex RI and O/C ratio (Nakayama et al., 2012; Lambe et al., 2013), others reported the opposite (Cappa et al., 2011; Nakayama et al., 2013). The spectral dependent data compiled by Moise et al., (2015) clearly indicate that measurements of the broadband optical properties of ambient





aerosols are scarce. This new retrieval approach is expected to contribute to future understanding of the optical properties of atmospheric aerosols.

## 4 Conclusions

We have developed a new approach to retrieve the time dependent broad spectrum optical properties of ambient aerosols

between 300 and 650 nm. Obtaining these properties over such a broad wavelength span and at a high time resolution can contribute considerably to our understanding of aerosol optical properties and their subsequent contribution to radiative forcing. Our approach is based on fitting and extrapolating instrumental extinction, scattering, and absorption data. In this study, the extinction coefficients were obtained using a homebuilt broadband cavity-enhanced spectrometer measuring at two distinct wavelength ranges: 315 to 345 nm and 360 to 390 nm. Extinction and absorption coefficients at $\lambda = 404$ nm

were measured using a homebuilt photo-acoustic spectrometer coupled to a cavity ring down spectrometer. The scattering coefficients at 457, 525, and 637 nm were measured using an integrating nephelometer. Although, the basic principles of the presented calculations may be used to represent ambient aerosol optical properties over any spectral range, depending on available instrumentation, it is important to investigate the weaknesses of the approach at spectral ranges beyond the instrumental wavelengths.

Computer simulations showed that, at the selected spectral range (300 to 650 nm) and for a wide atmospherically relevant range of effective complex refractive indices, the expected errors in the size weighted, wavelength- and time-dependent single scattering albedo, extinction and scattering coefficients, and in the real part of the effective complex refractive index are mostly less than ±10%. Although the relative errors in the imaginary part of the effective complex refractive index and in the wavelength- and time-dependent absorption coefficient are expected to grow with increasing

wavelength (as these parameters diminish), for a total column radiative transfer calculation, the corresponding absolute errors would be negligible. For example, under the conditions of this simulation, at 400 nm, the absolute errors on $\alpha_{abs}$ and $k$, are in the range of 1 to 3 Mm$^{-1}$ and 0.01 to 0.015, respectively.

Ambient measurements and measurements of the lab-generated brown carbon proxy aerosols demonstrated the effectiveness of our approach for laboratory, chamber, and ambient studies, where aerosol size selection may not be achieved

because of low number concentrations or a lack of sufficiently large particles. Retrieving the total distribution complex effective refractive index is a significantly faster method compared with size selective measurements, which are often used in order to derive aerosol refractive indices. It also minimizes possible errors arising from multiply charged particles (Miles et al., 2011) and from partial representation of the total size distribution due to limitations in the selectable size range. This study presents a first comparison between these two measurement approaches.

Application of the method to the continuous monitoring of ambient aerosols provides extensive and intensive time- and spectrally-dependent aerosol optical properties that may be applied in a variety of studies, such as investigations of the effect of chemical aging and SOA formation mechanisms or of hygroscopicity on the spectral dependency of optical properties. It





also emphasizes the sensitivity of the retrieval of the total distribution effective complex refractive index to fast changes in particle size distribution and concentration.

## 5 Acknowledgements

5  This research was partially funded by a grant from the German Israeli Science foundation (GIF), project no. 1136-26.8/2011 and by the USA-Israel Binational Science Foundation (BSF) grant no. 2012013. YR acknowledges support from the Dollond Charitable Trust. The SPARTAN station in Rehovot was supported by the Environmental Health Fund (grant # PGA 1402) and by the Weizmann Institute. NB acknowledges support from the Helen Kimmel Center for Planetary Sciences at the Weizmann Institute.

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



**Figure 1. Schematic of the photo-acoustic spectrometer (PAS) coupled to a cavity ring down (CRD) spectrometer (PA-CRD-S) and of the broadband cavity enhanced spectrometer (BBCES), with channels for 315 to 345 and 390 to 420 nm. The optical cavity of the CRD-S and the two optical cavities of the BBCES were assembled together in a rigid optical cage to minimize alignment stability issues. Abbreviations: CCD, charge coupled device; PBS, polarizing beam splitter; PD, photodiode; PMT, photomultiplier tube; TEC, thermoelectric cooler. The small black arrows indicate the entrance of the purge flows, and the thinker black arrows the direction of the aerosol flow.**



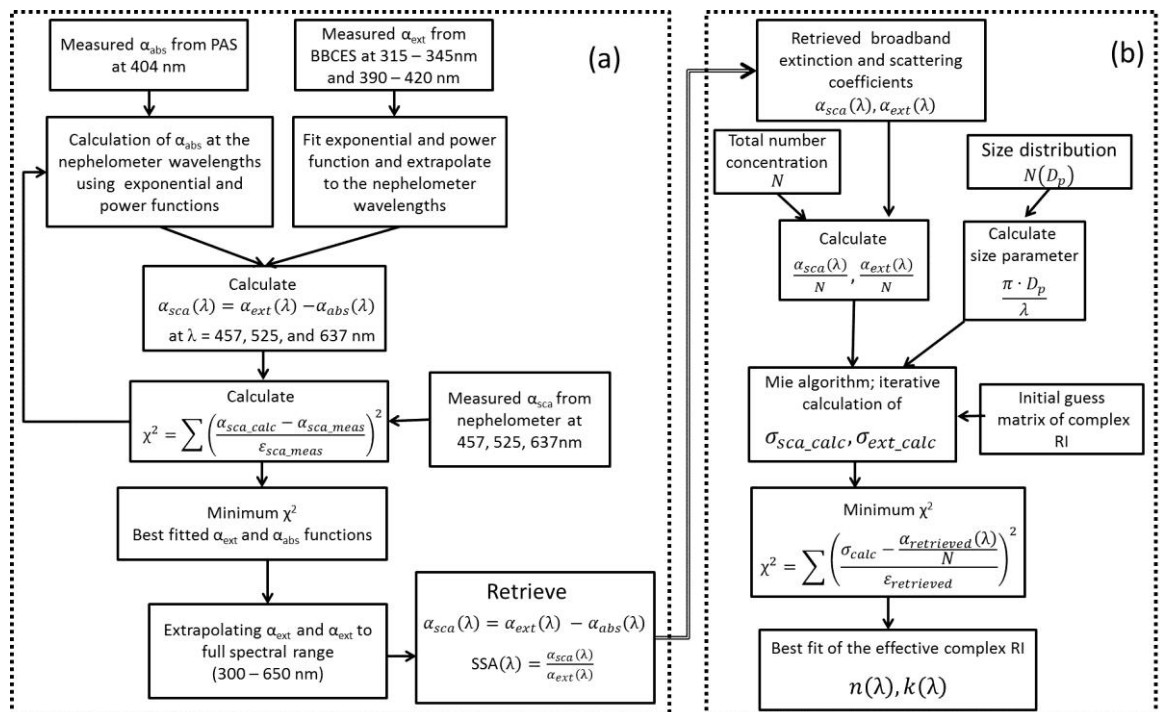

**Figure 2. (A)** Schematic of the broadband extinction, scattering, absorption, and single scattering albedo retrieval methodology. **(B)** Schematic of the method for retrieving the effective complex refractive index using the total particle size distribution. Abbreviations: $\alpha_{abs}$, $\alpha_{ext}$, and $\alpha_{sca}$, wavelength dependent absorption, extinction, and scattering coefficients, respectively; BBCES, broadband cavity enhanced spectrometer; the subscripts calc and meas indicate a calculated or measured value, respectively; $D_p$, particle diameter; $n$, the real part of the complex refractive index; $k$, the imaginary part of the complex refractive index; $N$, particle number concentration; PAS, photo-acoustic spectrometer; RI, effective complex refractive index; $\sigma_{ext\_calc}$ and $\sigma_{sca\_calc}$, theoretical extinction and scattering cross sections weighted by the size distribution; $\chi^2$, minimum square difference; $\omega$, size-weighted single scattering albedo (SSA).





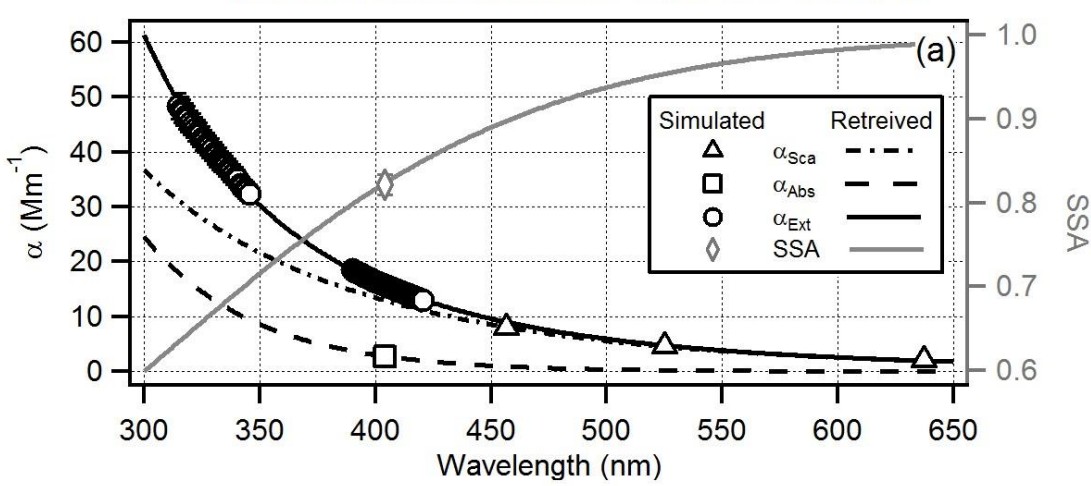

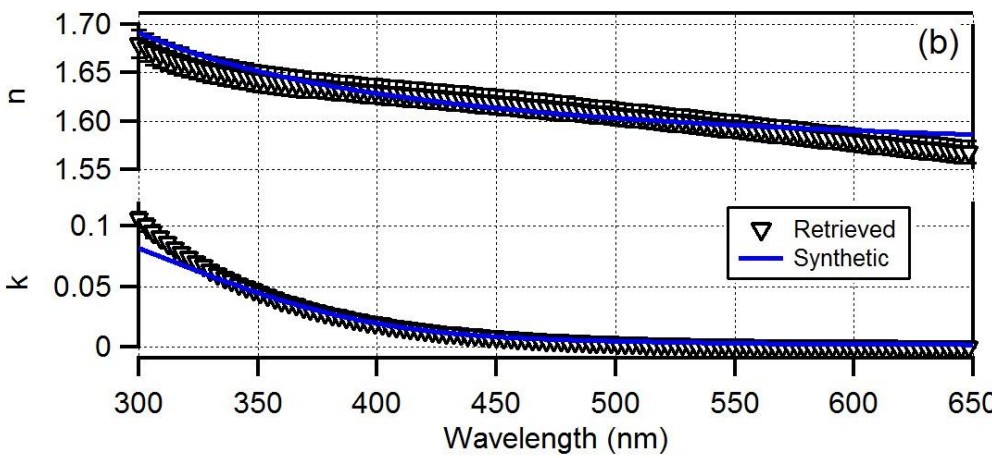

**Figure 3.** (a) Simulated wavelength-dependent extinction ($\alpha_{ext}$; circles), scattering ($\alpha_{sca}$; triangles), and absorption ($\alpha_{abs}$; square) coefficients from the synthetic complex refractive index (RI) shown in the lower panel (black lines). The retrieved broadband $\alpha_{ext}$ (black line), $\alpha_{sca}$ (dash-dot line), and $\alpha_{abs}$ (dashed line) and single scattering albedo (SSA; grey line and right axis) are also shown.

5 (b) The synthetic complex RI (black line) and the retrieved effective complex RI (circles), divided into their real and imaginary parts.



**Figure 4.** Left: schematic of the laboratory measurements. The nephelometer was position inside the laboratory and the aerosol went through it before being measured by the lab set up: optical system, PAS, CPC, and SMPS. Right: schematic of the ambient measurements. The nephelometer was moved and positioned on the roof. Abbreviations: DD, diffusion dryer; ISS, iso-kinetic splitter; CRD-S, cavity ring down spectrometer; BBCES, broadband cavity enhanced spectrometer; LPM, liters per minute; PAS, photoacoustic spectrometer; PM; particulate matter; CPC, condensation particle counter; SMPS, scanning mobility particle sizer.





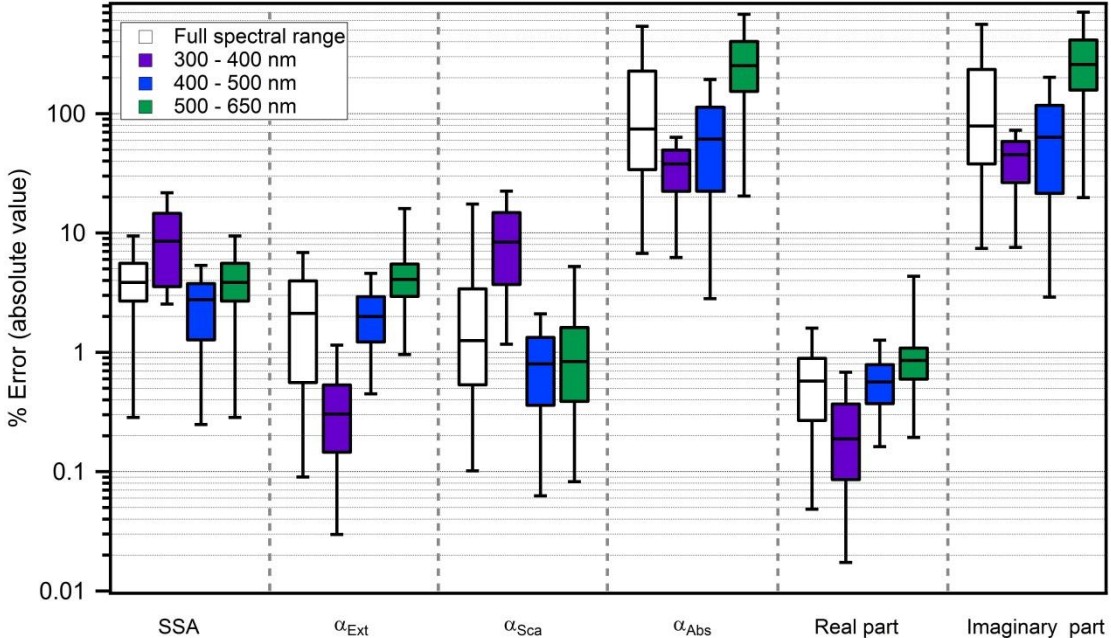

**Figure 5. Box plot of the absolute value percentage errors for the retrieved variables from the 100 different synthetic data sets of effective complex refractive indexes. The horizontal line within the box indicates the median, the boundaries of the box indicate the 25th and 75th percentile values, and the whiskers indicate the 5th and 95th percentile values of the results. Real and imaginary parts refer to the real and imaginary components of the refractive index. Abbreviations: SSA, size-weighted single scattering albedo; $\alpha_{abs}$, $\alpha_{ext}$, and $\alpha_{sca}$, wavelength dependent absorption, extinction, and scattering coefficients, respectively.**





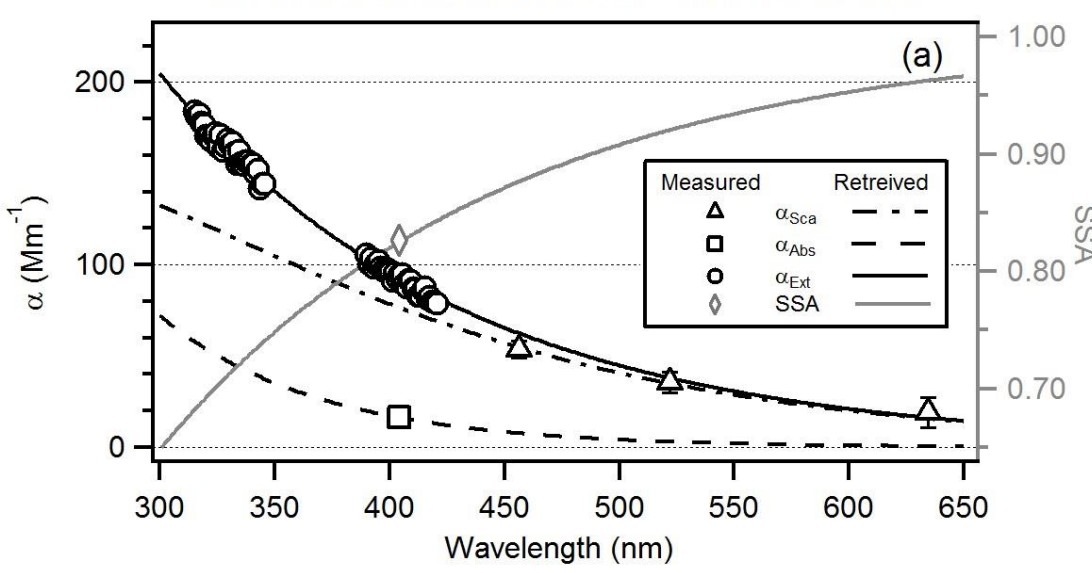

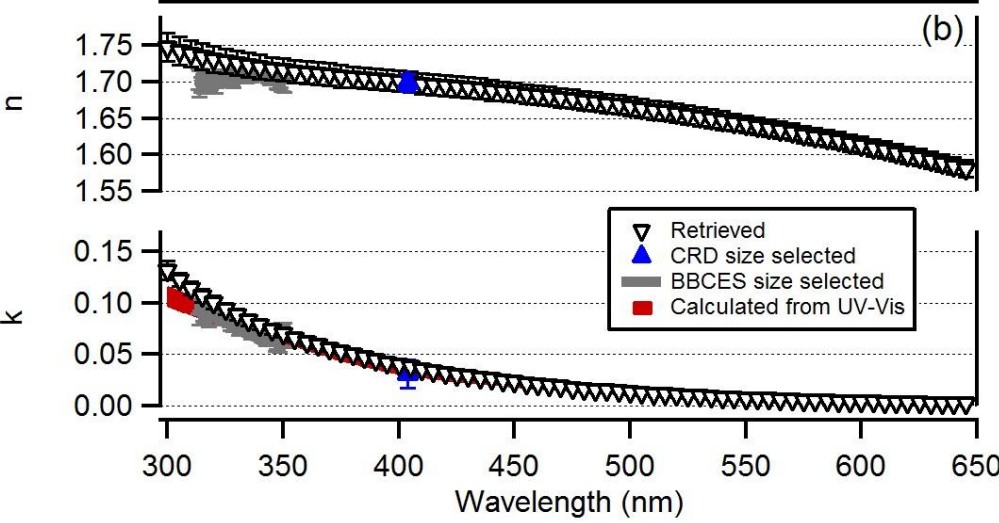

Figure 6. (a) Measured extinction (circles), scattering (triangles), and absorption (square) coefficients ($\alpha_{ext}$, $\alpha_{sca}$, and $\alpha_{abs}$, respectively) for Pahokee peat fulvic acid. The retrieved values for broadband extinction (black line), scattering (dash-dot line), absorption (dashed line) and single scattering albedo (SSA; grey line) are also shown. (b) Retrieved broadband complex refractive index for Pahokee peat fulvic acid using: 1) the retrieved RI from the data shown in panel (a) (inverted triangles); and 2) size selection measurements for the broadband cavity-enhanced spectrometer (BBCES-315; grey line) and the cavity ring down spectrometer (CRD-S) at 404 nm (blue triangles). The imaginary part of the refractive index calculated from UV-Vis absorption measurements is indicated by the red shaded area.





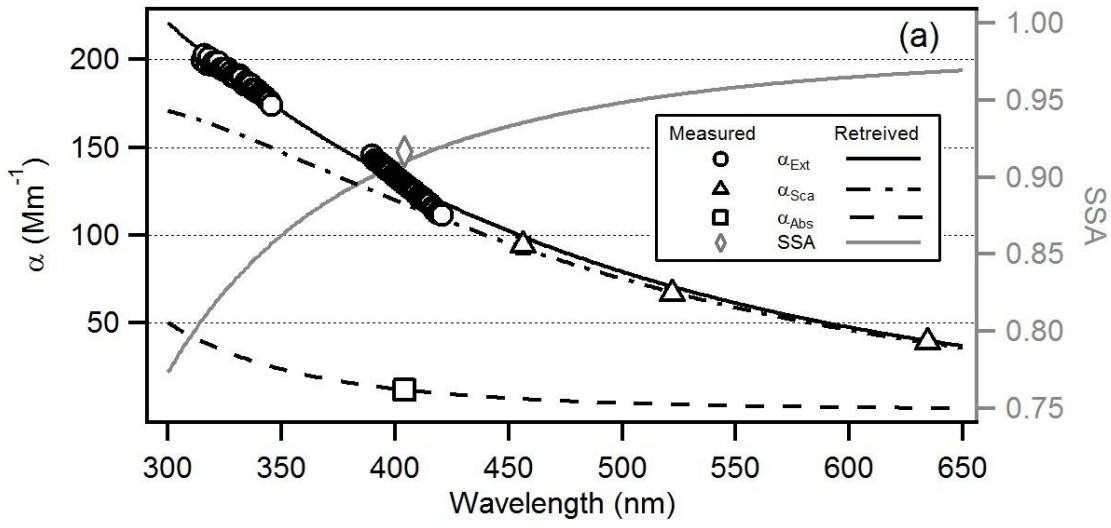

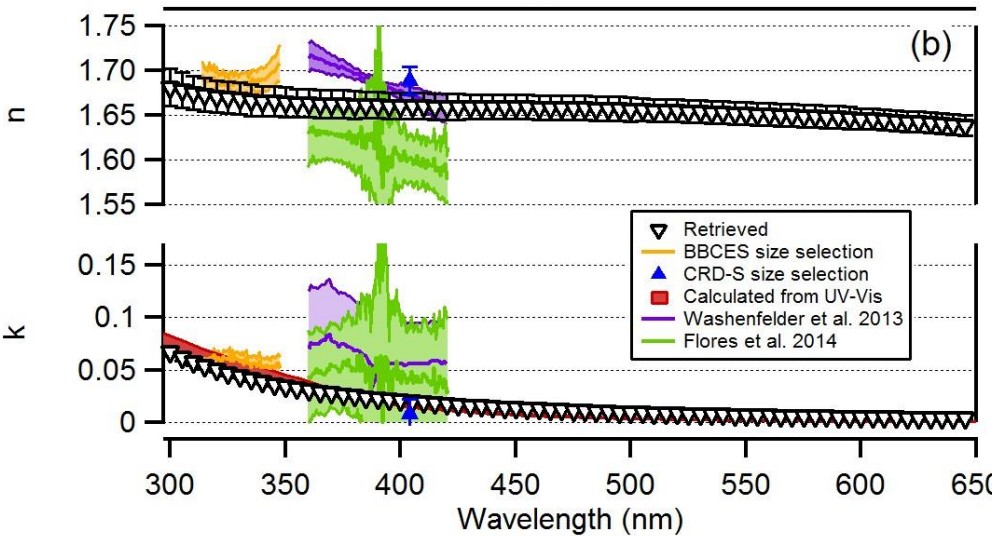

**Figure 7. (a) Measured extinction (circles), scattering (triangles) and absorption (square) coefficients ($\alpha_{ext}$, $\alpha_{sca}$, and $\alpha_{abs}$, respectively) for Suwannee river fulvic acid. The retrieved broadband extinction (black line), scattering (dash-dot line), absorption (dashed line) and single scattering albedo (SSA; grey line) are also shown. (b) Retrieved broadband complex refractive index (RI) for Suwannee river fulvic acid using: 1) the retrieved RI from the data shown in panel (a) (inverted triangles); (2) size selection measurements for the broadband cavity-enhanced spectrometer (BBCES-315; orange line) and the cavity ring down spectrometer (CRD-S) at 404nm (blue triangles); and (3) from the published data of Washenfelder et al. 2013 (purple line) and Flores et al. 2014 (green line). The imaginary part of the refractive index calculated from UV-Vis absorption measurements is indicated by the red shaded area.**





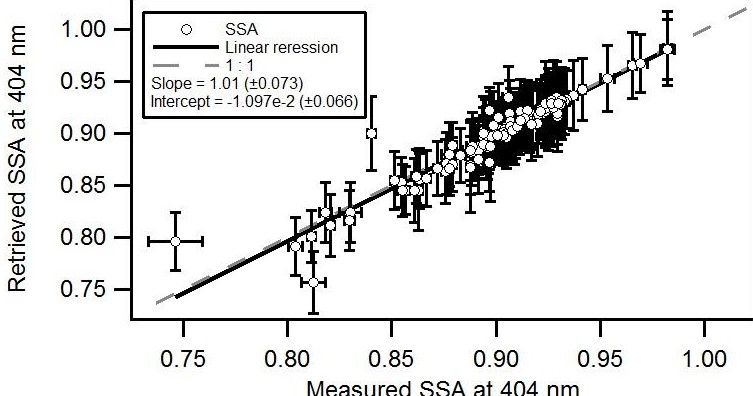

**Figure 8. Retrieved and measured single scattering albedo (*SSA*) at 404 nm wavelength. Retrieved *SSA* is calculated from the retrieved extinction and scatter coefficients ($\alpha_{ext}$(t) and $\alpha_{sca}$(t), respectively), while the measured *SSA* is calculated from the values of $\alpha_{ext}$(t) and $\alpha_{abs}$(t) obtained through direct measurement by the single wavelength photo acoustic spectrometer coupled to a**

5 **cavity ring down aerosol spectrometer (PA-CRD-S).**







**Figure 9. Time series of the retrieved coefficients for extinction (A), scattering (B), absorption (C), and of the single scattering albedo (SSA) (D) for the 300 to 650 nm wavelength range.**

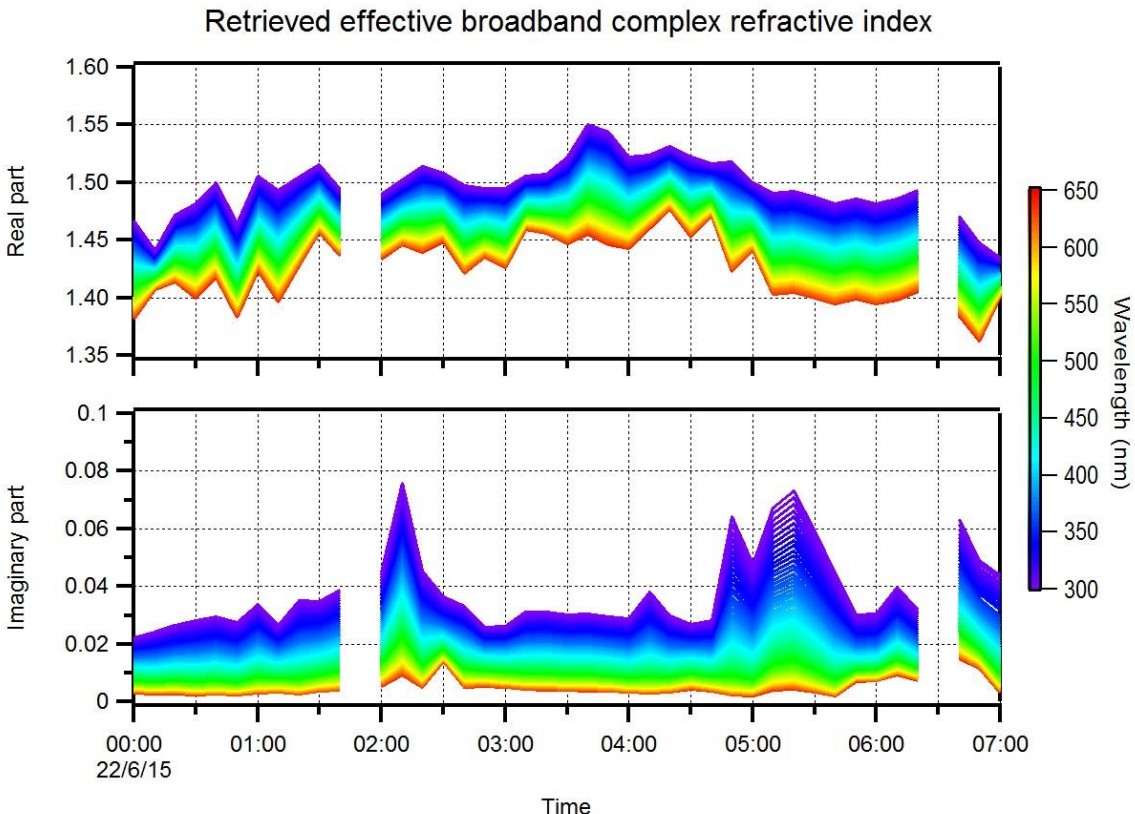

**Figure 10. Time series (night-time hours) of the real and imaginary components of the retrieved effective complex refractive index.**



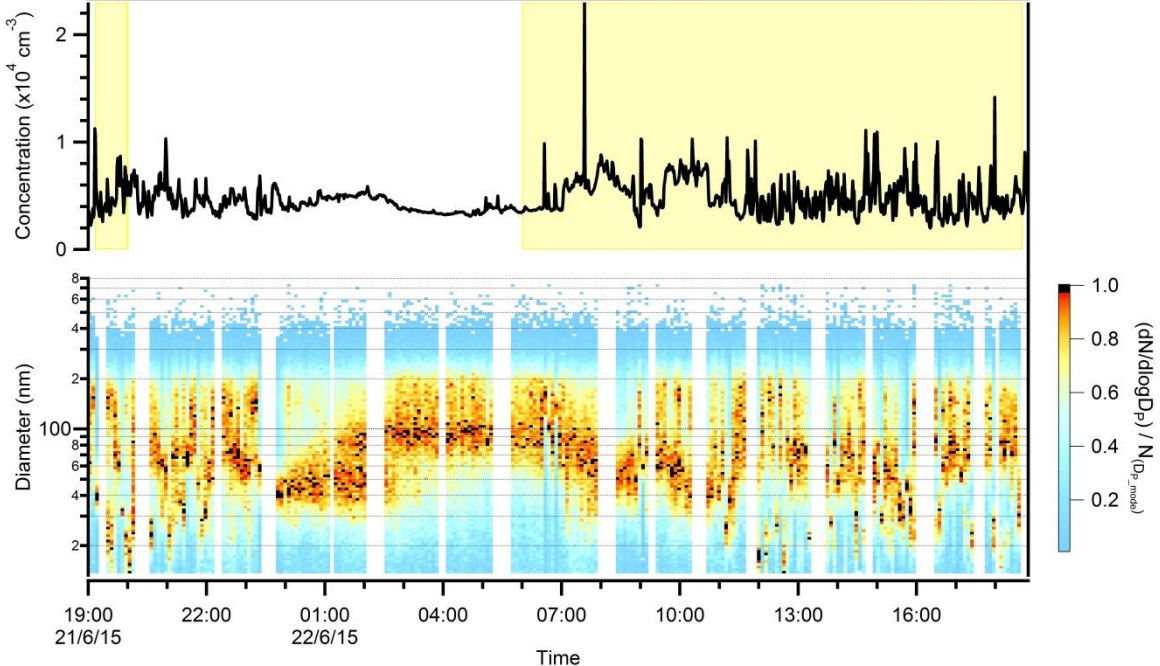

**Figure 11. Time series of the total number concentration (N; upper panel) and of the size distributions obtained from the scanning mobility particle sizer normalized to the mode diameter concentration (lower panel).**