# Peer review of "Correction of nephelometer data due to hygroscopic growth"

_Atmospheric Measurement Techniques, 2016_

## Referee Comment (RC1) · Anonymous Referee #3 · 26 Apr 2016

The manuscript describes the benefits of bringing together a number of complementary tools to measure the optical properties of ambient aerosol, with direct retrievals of the scattering, absorption and extinction coefficients, the single scattering albedo (SSA) and the complex refractive index. Measurements are made using two broadband cavity enhanced spectrometers operating over different wavelength ranges, a photoacoustic spectrometer with cavity ringdown, and a nephelometer. This combined approach could lead to significant improvements in measurements of the wavelength resolved scattering and absorption properties of ambient aerosol. The authors provide a substantial amount of data to benchmark the approach, comparing it with previous measurements. The paper is largely well-written and the Figures are clear. The authors should consider the following comments.

(1) The fitting procedure to get the absorption coefficient, the SSA and the imaginary

part of the RI depends on one direct absorption measurement, the PAS measurement at 405 nm which has a large error (stated to be typically 60 %). So, to get the values for these properties over the full wavelength range, the authors must assume a variety of power laws or exponential relationships with wavelength to relate the extinction, scattering and absorption measurements. This is very briefly described in section 2.3.1. This is a key part of the retrieval and I think more discussion and consideration of errors needs to be included here, a figure needs to be shown including the trial power laws etc. and the variation in the fitting error. This extra detail would be helpful to the reader.

(2) Although their retrievals broadly match their expectations for the trends (e.g. in figures 6 and 7), the impact of the errors on the SSA, single wavelength absorption measurements, extinction and scattering measurements on the retrievals need to be more fully discussed. For example, on these two figures, I would expect to see error envelopes on the lines shown in each of these plots, following from the cumulative effect of the uncertainties in the individual measurements. Even with the errors reported in section 3.1 from the simulations (of order 5, 10 and 60 %), I would expect the error envelopes to become large when outside the range of the measurements, for example in the retrieved value of the absorption coefficient below 400 nm in Figs. 6 and 7, particularly when the uncertainties in the nephelometer and PAS measurements are considered. These error envelopes should then be included in plots such as Figure 7(b) for the imaginary part of the refractive index. In this plot, error bars are shown in the real part but not the imaginary part, is there a reason for this? In Fig 7 there is a systematic error in the agreement between the retrieval of the extinction coefficient and the measured value around 400 nm with the BBCES data crossing the retrieved line – does this have consequences for the fit? To reduce errors, ideally there should be more overlap in wavelength between the measurements of scattering coefficient and extinction coefficient, what prospect is there for doing this? Depending on the errors on the imaginary part in Figs 6 and 7, can the authors now definitely state that their new measurements of the optical constants should be preferred over the previous measurements.

(3) I think the authors should stress in the abstract that continuous measurements of extinction coefficients are made between 315 and 345 nm and 390 to 420 nm, but other measurements are pointwise (404, 457, 525, 637 nm). The authors suggest that their approach gives optical coefficients and SSA over the full range 300 – 650 nm but this is not the measurement range. There does seem to be some confusion in the text over the upper wavelength range for the BBCES measurements: the upper range stated in 2.1 (360 to 390 nm) does not match with the range in 2.2.1 (390 to 420 nm). Also line 9 of page 11 says 360 to 390 nm.

(4) Line 7, page 7: for clarity, specify what the "individual wavelengths" are.

(5) Line 27, page 7: The real part of refractive index ranges from 1.692 at 300 nm (presumably this is the wavelength?) and 1.856 at 650 nm. These are much larger values that those expected for SOA, why have such large values been chosen? Why are values for imaginary parts of refractive index quoted to nearest 10-6 in magnitude – presumably the measurement is not this sensitive?

(6) Line 3, page 8: An additional error with normal distribution of +/- 2% standard deviation was added to simulated data set. Why this level of error?

(7) On page 10 line 20, the authors state "the retrieval of the effective complex RI is strongly dependent on accurate representation of the size distribution and aerosol particle number concentration." As a consequence they have only used data taken during the night when variations in number concentration are less severe. For the instrument to be robust in the field, is there a solution to this?

---

## Referee Comment (RC2) · Anonymous Referee #1 · 26 Apr 2016

General Comments

This manuscript describes a new approach for combining measurements of the extinction, scattering, and absorption by ambient aerosols with interpolation and extrapolation to span a spectral range of 300 nm to 650 nm. It also outlines a method for retrieving the effective complex index of refraction of the aerosols based on comparison of these measurements with Mie theory calculations. The utility of the approach lies in its ability to generate climate relevant optical properties across the UV-visible spectrum from measurements of polydisperse aerosol size populations at ambient concentrations. As such, it avoids complications and limitations associated with size selecting particles and opens up possibilities for monitoring spectral optical properties in real time and correlating them with observed chemical and physical changes. Overall, this is a well-written manuscript describing an advance in how aerosol optical properties

are measured and will be of interest to many in the field. It is recommended for publication in Aerosol Measurement and Techniques after the specific comments, below, are addressed.

Specific Comments

1. What are the detection limits for the photoacoustic spectrometer and the cavity ring down spectrometer?

2. There probably is not a lot of difference between using an exponential function and using a power law function to extrapolate absorption, scattering, and extinction (the authors can correct me if I am wrong). However, the power law function is more commonly, almost exclusively, used for all three measurements. Can the authors elaborate some on why they choose to employ combinations of both functions?

3. Related to point #2, assuming either an exponential or a power law function holds over the entire spectral range could introduce errors. For example, Massabo et al. [1] recently found that ambient aerosol absorption is better fit to a two- or three-power law function. Given that the current work extrapolates absorption from a single wavelength (404 nm), the choice of extrapolation function could have a sizable impact on the fitting and refractive index retrieval. It is worth noting that such an impact would not be obvious in the PPFA and SRFA samples since they have negligible black carbon components. Consequently, the error could impact the ambient measurements, though there are no independent measurements that could identify or constrain this potential error.

4. Figures 4 (schematic of sample flows) and 11 (SMPS plots) are not necessary and could be moved to supplemental information.

5. Figure 8 shows good agreement between calculated measured SSA at 404 nm, but both values use the measured extinction. A more direct and convincing demonstration of the agreement would be to compare calculated (alpha_ext(404) – alpha_sca(404))

and measured (alpha_abs(404)) absorption at 404 nm.

6. The manuscript would be strengthened significantly by including plots of the extinction/absorption Angstrom exponents, which could be calculated from the derived spectral values of extinction and absorption.

7. In the captions for Figures 9 and 10 it would be helpful to remind the reader that these ambient measurements are for the dry aerosol (i.e. do not include the water component which would alter substantially the optical properties). Also, the sampling location should be included in the captions.

8. To highlight the good agreement between the measured and calculated values, it would be interesting and illustrative to plot curves of the measured extinction (maybe one wavelength for each CES cell), scattering (457 nm, 525 nm, and 637 nm), absorption (404 nm), and SSA (404 nm) in Figures 9a, 9b, 9c, and 9d, respectively.

9. In the Conclusions it is confusing that the authors claim that such a large error on alpha_abs (404 nm) of 60% leads to only negligible errors in total column radiative transfer calculations. And, as the authors point out, the error is expected to be even larger at longer wavelengths where absorption is smaller. It seems like such large errors have to be important given that the absorption extrapolated from 404 nm is used to determine optimum agreement between the calculated and measured scattering values.

Technical Corrections

Page 1, line 1: "VU-Vis" should read "UV-Vis"

Page 1, lines 23 and 27: "EFR" should read "ERF" for consistency with other abbreviation ("ERFari")

Page 2, line 13: "white-type" should read "White-type"

Page 2, line 15: the detection limit for the White-type cells should be larger than that of

[Figure]

CES, not lower, since they have shorter effective path lengths

Page 3, line 30: it would seem that the colored glass filters

Page 4, line 28: "flown" should read "flowed"

Page 7, line 3: "Fig. 2b" should read "Fig. 2a"

References

(1) Massabò, D.; Caponi, L.; Bernardoni, V.; Bove, M. C.; Brotto, P.; Calzolai, G.; Cassola, F.; Chiari, M.; Fedi, M. E.; Fermo, P.; et al. Multi-Wavelength Optical Determination of Black and Brown Carbon in Atmospheric Aerosols. Atmos. Environ. 2015, 108, 1–12.

---

## Referee Comment (RC3) · Anonymous Referee #4 · 28 Apr 2016

Review of:

"A new approach for measuring the UV-Vis optical properties of ambient aerosols" by Bluvshtein et al.

This paper reports the method to estimate wavelength dependent optical properties (absorption, scattering, extinction, SSA, and refractive index) for aerosols by extrapolate and combined the observed data. The results is interesting and will be useful not only to determine the "effective refractive index" for ambient particles (assuming homogeneous spherical particles), but also the "refractive index" for POA and SOA generated in the laboratory. This manuscript includes sufficient originality, and the topic seems to fit the journal. I recommend publication to AMT after the points below have been addressed.

Major comments:

1) I think this proposed procedure to extrapolate the $\alpha_{abs}$ at 404 nm and $\alpha_{ext}$ at 315-345 nm and 390-450 nm to other wavelength is reasonable only when resonant wavelength of light absorption of particles exist at shorter wavelength compared to these measurement wavelengths. However, some types of SOAs are reported to have maximum light absorption at longer wavelengths. I recommend to adding some discussion on this issue.

2) Section 2.2.3

How did you calibrate the IN?

How did you estimate the truncation error of the IN?

The truncation error should depend on complex refractive index values.

3) Page 7, line 29

"Both $n(\lambda)$ and $k(\lambda)$ were scaled by two incoherent sine waves to simulate temporal variability."

Could you add some more explanation?

How is the amplitude of sine waves? Is it common method to simulate temporal variability? If so, pleas add a reference.

4) Page 8, line 3

The errors in $\alpha_{abs}$ for the calibration of PAS and the truncation correction for IN seem to be larger than 2%.

5) Page 8, line 17

"A complementary $N_2$ flow of 1.3 SLM was added and mixed with the sample flow, flow, which was then introduced to the IN and subsequently split equally to the SMPS, CPC, PAS, and a three-cavity optical cage that contained the CRD-S and BBCES (see Fig. 4)."

Why did you use the $N_2$ instead of air?

I think that the difference may influence to the measurements of PAS, IN, CRD-S, and BBCES.

6) Section 3.3 (Ambient aerosol measurement)

Many particles (e.g. black carbon (BC) and coated BC, dust) is not homogeneous sphere in the real atmosphere. Some description on this point may be needed.

7) Fig. 9(c)

The measured and retrieved $\alpha_{abs}$ at 300 nm are more than 10 times larger than $\alpha_{abs}$ at 600 nm throughout the day including the periods when traffic emission seems to have large contributions. Do you think the observed large wavelength dependence (or AAE) is due to the large contributions of brown carbon or inaccuracy of extrapolation of $\alpha_{abs}$ at 404 nm to 600 nm?

8) Supplemental material

The authors assumed negligible light absorption at 637 nm for ambient particles. I think it not common in urban atmosphere because of the existence of BC. I recommend to adding the validation of the assumption.

Minor comments:

1) Page 4, line 1

The value (0.99960) at 330 nm is not consistent with that in Fig.1.

2) Page 5, lines 18-19

"The laser power is continuously monitored and used to cancel variations in acoustic signal related to laser power fluctuations."

Where the laser power for the PAS is monitored in Fig. 1?

3) Page 7, line 28

I think the "1.856" should be "1.586".

4) Fig.2(a)

I think that one of $\alpha_{ext}$ in the bottom should be $\alpha_{abs}$.

---

## Referee Comment (RC4) · Anonymous Referee #5 · 2 May 2016

General comments:

This nicely written paper outlines the use of a combination of measurements of aerosol optical properties to retrieve continuous, spectrally dependent absorption, scattering, total extinction and complex refractive index over the atmospherically relevant range 300-650 nm. The novel combination of instruments and the novel retrieval algorithm make this paper easily suitable for AMT.

As the now fourth reviewer on this paper, I largely concur with the comments of the three reviews already posted, which will improve the presentation, and I will keep my own comments brief. I specifically endorse comments of referee #3 with respect to error analysis.

Specific Comments:

Page 6, line 24-26. The use of a power law for the alpha_abs seems specifically designed to capture brown carbon. What would the appropriate spectral dependence be for black carbon if that were the only absorbing component, and would the power low represent black carbon well in that situation?

Page 9, section 3.1 and Figure 5. A few questions / comments. 1) The relative errors are large in extinction due to absorption (and in imaginary refractive index) and small in extinction due to scattering and in total extinction (and real refractive index). One would guess that large errors in absorption would translate directly into large errors in SSA, which is directly proportional to absorption, yet errors in SSA are much smaller. Some explanation is warranted. 2) Relative errors are given with no indication of the sign of the errors. Does this analysis reveal any systematic deviation, or are the errors simply distributed about zero? 3) If I understand this procedure correctly, the measured values are assigned to the correct value in the synthetic data for the purpose of testing the retrieval. Please correct if I have misunderstood. What would be the effect and / or additional error in the retrieval if measurement uncertainty were considered (i.e., measurements assigned to values different from the synthetic data according to the measurement uncertainty distribution)?

Page 10, line 6-7: Wouldn't one expect agreement between the measurement and retrievals at the single wavelength to which the aerosol absorption is constrained in the retrieval? Suggest adding the phrase "as expected" or equivalent to indicate this.

---

## Author Comment (AC1) · 23 Jun 2016

1) What are the detection limits for the photoacoustic spectrometer and the cavity ring down spectrometer?

Our cavity ring down aerosol spectrometer (CRD-S) has the following properties: the length of the cavity between the mirrors is 0.95 m, the ratio between the cavity length and the length fill with aerosols (L/d) is 1.193. The typical empty cavity decay time ($\tau 0$) is 30 usec, exponential fitting residual at t=5*$\tau 0$ is less then +-10% (typically +-5%), $\tau 0$ standard deviation (std) is about 0.02% (+-0.006 usec). The limit of detection (LOD) defined as LOD = $\tau 0$+3*Std calculates to be 0.055 Mm-1, the limit of quantification (LOQ) defined as LOQ = $\tau 0$+10*Std is 0.185 Mm-1. The photoacoustic spectrometer base line is about 7.5 mV with std of 4.5 mV. Taking the average of 120 measurements

at 1 Hz sampling rate (2 minute long measurement) the standard error is 0.411 mV. With calibration slope of 48 V-1Mm-1 the LOD is about 0.06 Mm-1 and LOQ is about 0.2 Mm-1.

2) There probably is not a lot of difference between using an exponential function and using a power law function to extrapolate absorption, scattering, and extinction (the authors can correct me if I am wrong). However, the power law function is more commonly, almost exclusively, used for all three measurements. Can the authors elaborate some on why they choose to employ combinations of both functions?

For the same raw data, power law fitting would probably be slightly steeper than exponential decay function. It is correct that when fitting the same data, the difference between the two methods is not large, but often one approach provides a better fit than the other. From Mie theory calculation, power law dependency of the extinction and absorption with spectrum is expected for small particles (up to 100-300 nm diameters). For larger particles (and/or shorter wavelengths), the ripple structure of the Mie curve is expected to decrease the power law behavior of the spectral extinction and absorption curve. The exponential function is only intended to allow for increased quality of fitting.

3) Related to point #2, assuming either an exponential or a power law function holds over the entire spectral range could introduce errors. For example, Massabo et al. [1] recently found that ambient aerosol absorption is better fit to a two- or three-power law function. Given that the current work extrapolates absorption from a single wavelength (404 nm), the choice of extrapolation function could have a sizable impact on the fitting and refractive index retrieval. It is worth noting that such an impact would not be obvious in the PPFA and SRFA samples since they have negligible black carbon components. Consequently, the error could impact the ambient measurements, though there are no independent measurements that could identify or constrain this potential error.

Technically in this study we extrapolate the absorption from 4 wavelengths and not from one. The absorption measurement at 404 nm is used together with the scattering measurements at additional 3 wavelengths. The extrapolation technique and the error analysis are thoroughly discussed in sections 2.3.1 and 2.3.2. Additionally, black carbon is not expected to significantly increase the errors using the extrapolation approach since it is also treated with a power law behavior of the absorption and the extinction.

4) Figures 4 (schematic of sample flows) and 11 (SMPS plots) are not necessary and could be moved to supplemental information.

We accept the reviewer's suggestion. Figures 4 and 11 were moved to the supplementary material and are now referred to as figures S2 and S3.

5) Figure 8 shows good agreement between calculated measured SSA at 404 nm, but both values use the measured extinction. A more direct and convincing demonstration of the agreement would be to compare calculated (alpha_ext(404) – alpha_sca(404)) and measured (alpha_abs(404)) absorption at 404 nm.

We believe that the reviewer confused our results. We do not calculate both SSA values using the same extinction data points. The measured SSA is calculated from direct extinction and absorption measurements measured by the PA-CRD-S. The retrieved SSA is calculated from the extrapolated extinction curve that best fits the BBCES extinction measurement (315 to 345 nm and 390 to 420 nm) and the best fitted scattering curve. This is now explicitly explained in the caption of figure 7 that now reads: "Comparison between the retrieved and measured single scattering albedo (SSA) values at 404 nm. The retrieved SSA is calculated from the retrieved extinction and scattering coefficients ($\alpha$ext(t) and $\alpha$sca(t), respectively), while the measured SSA is calculated from the values of $\alpha$ext(t) and $\alpha$abs(t) obtained through direct measurement by the single wavelength photo acoustic spectrometer coupled to a cavity ring down aerosol spectrometer (PA-CRD-S)."

6) The manuscript would be strengthened significantly by including plots of the extinction/ absorption Angstrom exponents, which could be calculated from the derived spectral values of extinction and absorption.

A plot of extinction and absorption Angstrom coefficient was added to the main text as figure 9.

7) In the captions for Figures 9 and 10 it would be helpful to remind the reader that these ambient measurements are for the dry aerosol (i.e. do not include the water component which would alter substantially the optical properties). Also, the sampling location should be included in the captions.

We thank the Reviewer for these suggestions. Figure 9 (now, figure 8) caption now reads: "Figure 9. Time series of the retrieved coefficients for extinction (A), scattering (B), absorption (C), and of the single scattering albedo (SSA) (D) for the 300 to 650 nm wavelength range of dried ambient aerosols." Figure 10 caption now reads "Figure 10. Time series (night-time hours) of the real and imaginary components of the retrieved effective complex refractive index for the 300 to 650 nm wavelength range of dried ambient aerosols."

8) To highlight the good agreement between the measured and calculated values, it would be interesting and illustrative to plot curves of the measured extinction (maybe one wavelength for each CES cell), scattering (457 nm, 525 nm, and 637 nm), absorption (404 nm), and SSA (404 nm) in Figures 9a, 9b, 9c, and 9d, respectively.

We refrain from adding additional data to the retrieved optical coefficients figures because it would make these figures too loaded. We would also refrain from adding additional figures to the main text of this manuscript. We believe that figure 8 (retrieved Vs measured SSA at 404 nm, currently figure 7) is sufficient to show the good agreement between measured and calculated values. We did, however, add two additional figures to the supplementary material. One showing the good agreement of the retrieved and measured scattering coefficients at the nephelometer wavelengths (figure S4) and the other showing the agreement of the extinction coefficients at the center wavelengths of

the two BBCES cavities (figure S5).

9) In the Conclusions it is confusing that the authors claim that such a large error on alpha_abs (404 nm) of 60% leads to only negligible errors in total column radiative transfer calculations. And, as the authors point out, the error is expected to be even larger at longer wavelengths where absorption is smaller. It seems like such large errors have to be important given that the absorption extrapolated from 404 nm is used to determine optimum agreement between the calculated and measured scattering values.

In section 3.1. and in the conclusions the value of 60% error at 400 nm (+-60% error is the median error for 400 to 500 nm wavelength range) relates to error of the retrieved absorption coefficient relative to the synthetic data used for computer simulation. It is not an error or an uncertainty on measured absorption coefficients at 404 nm done with the PAS and used to extrapolate absorption data. For clarification, the relevant paragraph in section 3.1. now reads: "Under the conditions of this simulation at 400 nm wavelength (namely; the complex RI, particle size distribution, and number concentration), a relative error of 60% in retrieved values translates into absolute errors of 1 to 3 Mm-1 on $\alpha$abs and of 0.01 to 0.015 on k, respectively." The relevant conclusion paragraph now reads: "For example, under the conditions of this simulation, at 400 nm, the absolute errors on retrieved $\alpha$abs and k, are in the range of 1 to 3 Mm-1 and 0.01 to 0.015, respectively."

Technical Corrections We thank the Reviewer or the careful reading of the manuscript

10) Page 1, line 1: "VU-Vis" should read "UV-Vis". Corrected

11) Page 1, lines 23 and 27: "EFR" should read "ERF" for consistency with other abbreviation ("ERFari"). Corrected

12) Page 2, line 13: "white-type" should read "White-type". Corrected

13) Page 2, line 15: the detection limit for the White-type cells should be larger than

that of CES, not lower, since they have shorter effective path lengths. Corrected

14) Page 3, line 30: it would seem that the colored glass filters. this comment is not clear, possibly part of it is missing.

15) Page 4, line 28: "flown" should read "flowed" I think you meant page 5. Corrected

16) Page 7, line 3: "Fig. 2b" should read "Fig. 2a". Corrected

Please also note the supplement to this comment:
http://www.atmos-meas-tech-discuss.net/amt-2016-66/amt-2016-66-AC1-supplement.pdf
* * *
[Figure]

[Figure]

**Fig. 1.** Figure 9. Time series of the retrieved absorption and extinction Angstrom exponents (AAE ans EAE) for the 300 to 650 nm wavelength range.

[Figure]

**Fig. 2.** Figure S3. Retrieved and measured scattering coefficients at the nephelometer wavelengths (457, 525 and 637 nm).

**Fig. 3.** Figure S4. Retrieved and measured extinction coefficients at the center wavelengths of the two BBCES cavities (330 and 405 nm).

---

## Author Comment (AC2) · 23 Jun 2016

We would like to thank the Reviewer for his/her helpful remarks. Below, please find our detailed point by point replies to the comments made by the Reviewer.

1) The fitting procedure to get the absorption coefficient, the SSA and the imaginary part of the RI depends on one direct absorption measurement, the PAS measurement at 405 nm which has a large error (stated to be typically 60 %). So, to get the values for these properties over the full wavelength range, the authors must assume a variety of power laws or exponential relationships with wavelength to relate the extinction, scattering and absorption measurements. This is very briefly described in section 2.3.1. This is a key part of the retrieval and I think more discussion and consideration of errors needs to be included here, a figure needs to be shown including the trial power laws

etc. and the variation in the fitting error. This extra detail would be helpful to the reader.

The PAS measurement error at 404 nm was not stated at 60% error (this was addressed in response to comment 9 of reviewer #1). In fact, Lack et al., (2012) stated that the instruments uncertainty is probably lower than 5%. We acknowledge the fact that the discussion of uncertainty analysis was not clear enough. To address this issue, sections 2.3.1 and 2.3.2 were revised and the discussions is now more elaborated and includes more details relating to the uncertainty propagation of the retrieval.

2) Although their retrievals broadly match their expectations for the trends (e.g. in figures 6 and 7), the impact of the errors on the SSA, single wavelength absorption measurements, extinction and scattering measurements on the retrievals need to be more fully discussed. For example, on these two figures, I would expect to see error envelopes on the lines shown in each of these plots, following from the cumulative effect of the uncertainties in the individual measurements.

To address the issue of uncertainty propagation in the retrieval procedure, sections 2.3.1 and 2.3.2 were revised and they now include more details . The calculated uncertainty envelops were added to figures 6a and 7a (now 5a and 6a).

Even with the errors reported in section 3.1 from the simulations (of order 5, 10 and 60 %), I would expect the error envelopes to become large when outside the range of the measurements, for example in the retrieved value of the absorption coefficient below 400 nm in Figs. 6 and 7, particularly when the uncertainties in the nephelometer and PAS measurements are considered.

As for the possibility of increased uncertainty outside the range of measurement, we should mention that figures 6a and 7a (now 5a and 6a) show the retrieved optical coefficients which are the fitted curves. As such the uncertainty on each individual point (wavelength) depends on the uncertainty of the fitted coefficients and not on the distance from the measured data point.

These error envelopes should then be included in plots such as Figure 7(b) for the imaginary part of the refractive index. In this plot, error bars are shown in the real part but not the imaginary part, is there a reason for this?

Uncertainties are also included in figures 6b and 7b (the retrieved complex RI) (now 5b and 6b) as error bars, there were simply too small to see with the size of the symbol. The two figures were changed to show the error bars more clearly.

In Fig 7 there is a systematic error in the agreement between the retrieval of the extinction coefficient and the measured value around 400 nm with the BBCES data crossing the retrieved line– does this have consequences for the fit? To reduce errors, ideally there should be more overlap in wavelength between the measurements of scattering coefficient and extinction coefficient, what prospect is there for doing this?

We completely agree with the reviewer's perspective on this matter. This could also allow for direct absorption calculation in wavelengths outside the current PAS measurement wavelength. Unfortunately current available nephelometers provide measurements at three wavelengths at the most and our CES instruments were dedicated to measure extinction at lower wavelengths were brown carbon light absorption maybe significant. However, we show here conceptually how such measurements can be done and with availability of instrumentation, this method can be implemented to deduce retrievals with lower errors.

Depending on the errors on the imaginary part in Figs 6 and 7, can the authors now definitely state that their new measurements of the optical constants should be preferred over the previous measurements. Previous studies measured only extinction and it has been shown that adding additional direct measurements of relevant properties such as absorption or scattering improves the accuracy of the retrieval (Zarzana et al., 2014). Additionally, there is an agreement between our broad band retrieval and the UV-Vis retrieval.

3) I think the authors should stress in the abstract that continuous measurements of

extinction coefficients are made between 315 and 345 nm and 390 to 420 nm, but other measurements are pointwise (404, 457, 525, 637 nm). The authors suggest that their approach gives optical coefficients and SSA over the full range 300 – 650 nm but this is not the measurement range. There does seem to be some confusion in the text over the upper wavelength range for the BBCES measurements: the upper range stated in 2.1 (360 to 390 nm) does not match with the range in 2.2.1 (390 to 420 nm). Also line 9 of page 11 says 360 to 390 nm.

The abstract was revised and it now addresses the reviewer's concerns. Additionally, some typos regarding the BBCES measurement range were corrected throughout the manuscript.

4) Line 7, page 7: for clarity, specify what the "individual wavelengths" are.

This line was revised to: "the effective complex RI of the total particle size distribution is retrieved at each individual wavelength (300 to 650 nm)".

5) Line 27, page 7: The real part of refractive index ranges from 1.692 at 300 nm (presumably this is the wavelength?) and 1.856 at 650 nm. These are much larger values that those expected for SOA, why have such large values been chosen? Why are values for imaginary parts of refractive index quoted to nearest 10-6 in magnitude – presumably the measurement is not this sensitive?

We thank the Reviewer for this point. The value of 1.856 is a typo, and it is now revised to 1.586. The real part values chosen for the simulation are within current knowledge (Moise et al., 2015). In this manuscript, imaginary parts of the refractive indices are quoted in a magnitude of 10-3, not 10-6.

6) Line 3, page 8: An additional error with normal distribution of +/- 2% standard deviation was added to simulated data set. Why this level of error?

To address the Reviewer's concern, the following sentence was added: "This value of error represents typical uncertainty values associated with our instrumental measurements."

7) On page 10 line 20, the authors state "the retrieval of the effective complex RI is strongly dependent on accurate representation of the size distribution and aerosol particle number concentration." As a consequence they have only used data taken during the night when variations in number concentration are less severe. For the instrument to be robust in the field, is there a solution to this? This is a good point.

This can be done by adding a large residence time volume to damp aerosol concentration variation to correspond to the measurement time of the slowest instrument, in this case the SMPS scan time. The following sentence was added: "In field applications, a large volume with long residence time of the sampled air can be added to the system to reduce variations in aerosol concentration."

Lack, D. A., Richardson, M. S., Law, D., Langridge, J. M., Cappa, C. D., McLaughlin, R. J., and Murphy, D. M.: Aircraft instrument for comprehensive characterization of aerosol optical properties, part 2: Black and brown carbon absorption and absorption enhancement measured with photo acoustic spectroscopy, Aerosol Sci Tech, 46, 555-568, 2012.

Moise, T., Flores, J. M., and Rudich, Y.: Optical properties of secondary organic aerosols and their changes by chemical processes, Chemical reviews, 115, 4400-4439, 2015.

Zarzana, K. J., Cappa, C. D., and Tolbert, M. A.: Sensitivity of aerosol refractive index retrievals using optical spectroscopy, Aerosol Sci Tech, 48, 1133-1144, 2014.

Please also note the supplement to this comment:
http://www.atmos-meas-tech-discuss.net/amt-2016-66/amt-2016-66-AC2-supplement.pdf

[Figure]

[Figure]

[Figure]

**Fig. 1.** Figure 5. (a) Measured extinction (circles), scattering (triangles), and absorption (square) coefficients ($\alpha$ext, $\alpha$sca, and $\alpha$abs, respectively) for Pahokee peat fulvic acid (error bars represent measur

[Figure]

[Figure]

**Fig. 2.** Figure 6. (a) Measured extinction (circles), scattering (triangles) and absorption (square) coefficients (αext, αsca, and αabs, respectively) for Suwannee river fulvic acid (error bars representing me

---

## Author Comment (AC3) · 23 Jun 2016

We would like to thank the Reviewer for his/her helpful remarks. Below, please find our detailed point by point replies to the comments made by the Reviewer.

1) I think this proposed procedure to extrapolate the $\alpha$abs at 404 nm and $\alpha$ext at 315-345 nm and 390-450 nm to other wavelength is reasonable only when resonant wavelength of light absorption of particles exist at shorter wavelength compared to these measurement wavelengths. However, some types of SOAs are reported to have maximum light absorption at longer wavelengths. I recommend to adding some discussion on this issue.

We acknowledge the importance of this distinction. The following paragraph was added in the introduction section: "The interaction of atmospheric fine particulate matter with

sun light was shown to resemble a power law dependence on wavelength decades ago (Ångström, 1929, 1930, 1961, 1964). The Ångström exponent was since widely used to describe this wavelength dependency and to characterize aerosols size (Valenzuela et al., 2015), composition (Russell et al., 2010) and source (Garg et al., 2016). This monotonic increase in extinction (scattering and absorption) with increasing particle size or with decreasing wavelength is observed for particles with radii smaller than the incident wavelength (or size parameter $< 2\pi$). For larger particles (or shorter wavelengths) the ripple and interference structures of the Mie curves significantly reduce the monotonic increase pattern (Bohren and Huffman, 1983). This also depends on the complex refractive index. Increasing real part limits the power low behavior to smaller particles while increasing imaginary part, dampening the ripple and interference structures, allowing power low behavior for larger particles. Particulate matter with molecular absorption bands in the actinic flux regain would also deviate from the power law spectral dependency. In an ambient dust-free atmosphere as well as in many laboratory and chamber experiments, particles are rarely larger than several hundreds of nanometers making the power law wavelength dependency assumption a resendable one (Kirchstetter et al., 2004; Hoffer et al., 2006; Sun et al., 2007; Chen and Bond, 2010; Washenfelder et al., 2015)."

2) Section 2.2.3: How did you calibrate the IN? How did you estimate the truncation error of the IN? The truncation error should depend on complex refractive index values.

The IN was calibrated by flowing $CO_2$ and $N_2$ and relating the measured signal to literature values of the Rayleigh scattering cross sections(Thalman et al., 2014). The truncation angle of the instrument is reported by the manufacturer to be 7-170 degrees (Snider et al., 2015). The truncation error is mostly dependent on the scatterer size and on the scattered wavelength or the particles size parameter. Mie theory calculation showed that for relatively small particles (up to about 300 nm diameter) the ratio of light intensity scattered at 7-170 degrees to the total scattered light is similar (or not significantly different) to that of Rayleigh scattering regardless of refractive index. In
this case the calibration procedure using gasses with known Rayleigh scattering co-efficients corrects for the truncation error. When sampling ambient aerosols, mostly composed of primary emissions and SOA with negligible dust component, the concentration of particles larger than 300 nm is negligible. For more details on integrating nephelometer truncation error please refer to Anderson et al., (1996).

3) Page 7, line 29: "Both $n(\lambda)$ and $k(\lambda)$ were scaled by two incoherent sine waves to simulate temporal variability." Could you add some more explanation? How is the amplitude of sine waves? Is it common method to simulate temporal variability? If so, pleas add a reference. The reviewer's question regarding the sine wave amplitude is not clear. The sentence was revised to include the sine waves amplitudes: "Both $n(\lambda)$ and $k(\lambda)$ were scaled by two incoherent sine waves to simulate temporal variability. The sine wave amplitude ranged from 1 to 1.05 for $n(\lambda)$ and from 1 to 1.1 for $k(\lambda)$."

4) Page 8, line 3: The errors in $\alpha$abs for the calibration of PAS and the truncation correction for IN seem to be larger than 2%.

The uncertainty in absorption coefficient measured with this PAS instrument as a result of the calibration procedure was estimated by Lack et al., (2012) to be less then $\pm 5\%$. The uncertainty in extinction cross section for the BBCES cavities was reported by Washenfelder et al., (2013) to be 3.6-4.1%. With the two largest contributors being the CPC particle counting and the cavity length to filled length ratio (RL). Both of which are not included in our simulation because we are producing extinction coefficients and not cross sections, and our cavities are built with a central inlet and two sides outlets which reduces the uncertainty in RL to the uncertainty in length measurement which is about 0.5 mm out of the full cavity length i.e. less than 0.1% . The uncertainty associated with the calibration procedure of the IN is based on uncertainty in temperature and pressure measurements and uncertainty in literature values of Rayleigh scattering cross sections of N2 and CO2. All of which are less than 2% (Thalman et al., 2014). The uncertainty associated to truncation errors of particles that is significantly different from that of the calibration gasses is considered negligible for small particles (Anderson et

al., 1996) and is not likely to be larger than 2%. The error we assigned in the simulation is normally distributed with standard deviation of 2%. This means that 68% of the optical coefficients are assigned with errors of up to $\pm$2%, 27% of the optical coefficients are assigned with errors between $\pm$2% and $\pm$4% and 4% of the optical coefficients are assigned with errors between $\pm$4% and $\pm$6%. Eventually, based on literature and personal experience we believe that our error assignment is not unreasonable.

5) Page 8, line 17: "A complementary N2 flow of 1.3 SLM was added and mixed with the sample flow, which was then introduced to the IN and subsequently split equally to the SMPS, CPC, PAS, and a three-cavity optical cage that contained the CRD-S and BBCES (see Fig. 4)." Why did you use the N2 instead of air? I think that the difference may influence to the measurements of PAS, IN, CRD-S, and BBCES.

We respectfully disagree with the reviewer. In the laboratory experiments particles are generated using a particle free dry nitrogen gas (described in the text), so additional nitrogen dilution would not change carrier gas properties.

6) Section 3.3 (Ambient aerosol measurement). Many particles (e.g. black carbon (BC) and coated BC, dust) is not homogeneous sphere in the real atmosphere. Some description on this point may be needed.

Correct, this is why we defined in the text: "The term effective complex RI is used to represent the whole particle size distribution. It is the complex RI from which, for the corresponding size distribution, we derive the optical coefficients that agree most closely with the measured or input values." This does not mean that we assume anything regarding the shape, composition and mixing state of the ambient particles.

7) Fig. 9(c) The measured and retrieved $\alpha$abs at 300 nm are more than 10 times larger than $\alpha$abs at 600 nm throughout the day including the periods when traffic emission seems to have large contributions. Do you think the observed large wavelength dependence (or AAE) is due to the large contributions of brown carbon or inaccuracy of extrapolation of $\alpha$abs at 404 nm to 600 nm?

A factor of 10 increment in the absorption from 650 nm to 300 nm wavelength would suggest absorption angstrom exponent of about 2.9. Although this is a high value for atmospheric aerosols, values such as these and larger have been reported previously (Hoffer et al., 2004; Kirchstetter et al., 2004; Kaskaoutis et al., 2007; Sun et al., 2007; Chen and Bond, 2010; Russell et al., 2010; Lack et al., 2012; Backman et al., 2014) for BrC aerosols and for desert dust containing aerosol population (which is resendable for east Mediterranean basin during spring). The AAE or the ratio between absorption co-efficients at 300 and 650 nm are mostly dependent on the wavelength dependency of the imaginary part of the complex refractive index (RI). Consider the following theoretical example for a single spherical BrC particle with diameter of 80 nm and the effective RI described in the figure. In this example $k300/k650=7.75$ and $Abs300/Abs650=20.5$, notice that this theoretical particle is hardly absorbing. We acknowledge the fact that black carbon contribution is expected to reduce the wavelength dependency in an urban environment. The reviewer's comment led us to find and correct a minor bug in the nephelometer data correction algorithm. Re-analysis of the measurements showed a stronger decrease in the absorption wavelength dependency (AAE) during the morning hours when higher BC contribution is expected. As can be seen in figures 9c and 9d (now 8c and 8d) there is a general increase in absorption at the longer wavelengths between 07:00 and 10:00 am but it is possible that the nature and distance of the sub-urban to urban sampling site from the main road reduced the black carbon contribution. With the lack of direct ambient aerosols absorption measurements at such wavelengths this question remains an open one.

8) Supplemental material. The authors assumed negligible light absorption at 637 nm for ambient particles. I think it not common in urban atmosphere because of the existence of BC. I recommend to adding the validation of the assumption.

Unfortunately we cannot show validation of this because we cannot directly measure absorption or extinction minus scattering at 637 nm wavelength. Although this is a very important issue for ambient measurement and for the nephelometer hygroscopic

growth correction procedure, it is beside the point for the applicability and robustness of the broadband extrapolation and retrieval procedure because ideally all instrumentation measure the same aerosol population dried or not, as was shown in the laboratory application. Additionally, the fitting procedure of the absorption curve, as described in section 2.3.1 allows for non-zero absorption even at 650 nm. This is also clear from figure 9, showing non-zero retrieved absorption at 650 nm.

Minor comments: 9) Page 4, line 1 The value (0.99960) at 330 nm is not consistent with that in Fig.1. Thank you. Corrected.

10) Page 5, lines 18-19 "The laser power is continuously monitored and used to cancel variations in acoustic signal related to laser power fluctuations." Where the laser power for the PAS is monitored in Fig. 1?

This was changed to "The laser power is continuously monitored using a photodiode at the back side mirror, and used to cancel variations in acoustic signal related to laser power fluctuations."

11) Page 7, line 28 I think the "1.856" should be "1.586". Revised

12) Fig.2(a) I think that one of $\alpha$ext in the bottom should be $\alpha$abs. Revised

Anderson, T. L., Covert, D. S., Marshall, S. F., Laucks, M. L., Charlson, R. J., Waggoner, A. P., Ogren, J. A., Caldow, R., Holm, R. L., Quant, F. R., Sem, G. J., Wiedensohler, A., Ahlquist, N. A., and Bates, T. S.: Performance characteristics of a high-sensitivity, three-wavelength, total scatter/backscatter nephelometer, J Atmos Ocean Tech, 13, 967-986, 1996.

Ångström, A.: On the atmospheric transmission of sun radiation and on dust in the air, 11, 156-166, 1929.

Ångström, A.: On the atmospheric transmission of sun radiation. Ii, 12, 130-159, 1930.

Ångström, A.: Techniques of determinig the turbidity of the atmosphere1, 13, 214-223,

1961.

Ångström, A.: The parameters of atmospheric turbidity, 16, 64-75, 1964.

Backman, J., Virkkula, A., Vakkari, V., Beukes, J. P., Van Zyl, P. G., Josipovic, M., Piketh, S., Tiitta, P., Chiloane, K., Petaja, T., Kulmala, M., and Laakso, L.: Differences in aerosol absorption angstrom exponents between correction algorithms for a particle soot absorption photometer measured on the south african highveld, Atmos Meas Tech, 7, 4285-4298, 2014.

Bohren, C. F., and Huffman, D. R.: Absorption and scattering of light by small particles, John Wiley & sons, INC, United States of America, 530 pp., 1983.

Chen, Y., and Bond, T. C.: Light absorption by organic carbon from wood combustion, Atmos. Chem. Phys., 10, 1773-1787, 2010.

Garg, S., Chandra, B. P., Sinha, V., Sarda-Esteve, R., Gros, V., and Sinha, B.: Limitation of the use of the absorption angstrom exponent for source apportionment of equivalent black carbon: A case study from the north west indo-gangetic plain, Environmental science & technology, 50, 814-824, 2016.

Hoffer, A., Kiss, G., Blazso, M., and Gelencser, A.: Chemical characterization of humic-like substances (hulis) formed from a lignin-type precursor in model cloud water, Geophys. Res. Lett., 31, 2004.

Hoffer, A., Gelencser, A., Guyon, P., Kiss, G., Schmid, O., Frank, G. P., Artaxo, P., and Andreae, M. O.: Optical properties of humic-like substances (hulis) in biomass-burning aerosols, Atmos. Chem. Phys., 6, 3563-3570, 2006.

Kaskaoutis, D. G., Kambezidis, H. D., Hatzianastassiou, N., Kosmopoulos, P. G., and Badarinath, K. V. S.: Aerosol climatology: Dependence of the angstrom exponent on wavelength over four aeronet sites, 2007, 7347-7397, 2007.

Kirchstetter, T. W., Novakov, T., and Hobbs, P. V.: Evidence that the spectral depen-

dence of light absorption by aerosols is affected by organic carbon, J Geophys Res-Atmos, 109, n/a-n/a, 2004.

Lack, D. A., Richardson, M. S., Law, D., Langridge, J. M., Cappa, C. D., McLaughlin, R. J., and Murphy, D. M.: Aircraft instrument for comprehensive characterization of aerosol optical properties, part 2: Black and brown carbon absorption and absorption enhancement measured with photo acoustic spectroscopy, Aerosol Sci Tech, 46, 555-568, 2012.

Russell, P. B., Bergstrom, R. W., Shinozuka, Y., Clarke, A. D., DeCarlo, P. F., Jimenez, J. L., Livingston, J. M., Redemann, J., Dubovik, O., and Strawa, A.: Absorption angstrom exponent in aeronet and related data as an indicator of aerosol composition, Atmos. Chem. Phys., 10, 1155-1169, 2010.

Snider, G., Weagle, C. L., Martin, R. V., van Donkelaar, A., Conrad, K., Cunningham, D., Gordon, C., Zwicker, M., Akoshile, C., Artaxo, P., Anh, N. X., Brook, J., Dong, J., Garland, R. M., Greenwald, R., Griffith, D., He, K., Holben, B. N., Kahn, R., Koren, I., Lagrosas, N., Lestari, P., Ma, Z., Martins, J. V., Quel, E. J., Rudich, Y., Salam, A., Tripathi, S. N., Yu, C., Zhang, Q., Zhang, Y., Brauer, M., Cohen, A., Gibson, M. D., and Liu, Y.: Spartan: A global network to evaluate and enhance satellite-based estimates of ground-level particulate matter for global health applications, Atmos Meas Tech, 8, 505-521, 2015.

Sun, H. L., Biedermann, L., and Bond, T. C.: Color of brown carbon: A model for ultraviolet and visible light absorption by organic carbon aerosol, Geophys. Res. Lett., 34, 2007.

Thalman, R., Zarzana, K. J., Tolbert, M. A., and Volkamer, R.: Rayleigh scattering cross-section measurements of nitrogen, argon, oxygen and air, J Quant Spectrosc Ra, 147, 171-177, 2014.

Valenzuela, A., Olmo, F. J., Lyamani, H., Anton, M., Titos, G., Cazorla, A., and Alados-

Arboledas, L.: Aerosol scattering and absorption angstrom exponents as indicators of dust and dust-free days over granada (spain), Atmos. Res., 154, 1-13, 2015.

Washenfelder, R. A., Flores, J. M., Brock, C. A., Brown, S. S., and Rudich, Y.: Broadband measurements of aerosol extinction in the ultraviolet spectral region, Atmos Meas Tech, 6, 861-877, 2013.

Washenfelder, R. A., Attwood, A. R., Brock, C. A., Guo, H., Xu, L., Weber, R. J., Ng, N. L., Allen, H. M., Ayres, B. R., Baumann, K., Cohen, R. C., Draper, D. C., Duffey, K. C., Edgerton, E., Fry, J. L., Hu, W. W., Jimenez, J. L., Palm, B. B., Romer, P., Stone, E. A., Wooldridge, P. J., and Brown, S. S.: Biomass burning dominates brown carbon absorption in the rural southeastern united states, Geophys. Res. Lett., 42, 653-664, 2015.

Please also note the supplement to this comment:
http://www.atmos-meas-tech-discuss.net/amt-2016-66/amt-2016-66-AC3-supplement.pdf

---

## Author Comment (AC4) · 23 Jun 2016

a figure relating to comment number 7.

———————————————————

[Figure]

**Fig. 1.**

---

## Author Comment (AC5) · 23 Jun 2016

[yinon.rudich@weizmann.ac.il](mailto:yinon.rudich@weizmann.ac.il)

Received and published: 23 June 2016

We would like to thank the Reviewer for his/her helpful remarks. Below, please find our detailed point by point replies to the comments made by the Reviewer.

Page 6, line 24-26. The use of a power law for the  $\alpha_{abs}$  seems specifically designed to capture brown carbon. What would the appropriate spectral dependence be for black carbon if that were the only absorbing component, and would the power law represent black carbon well in that situation?

Pure black carbon was shown (Moosmuller et al., 2011) to have a distinct power law behavior of absorption with AAE of 1. However, there is no inherent limitation in the presented approach that would prevent fitting of a power law with AAE of 1 or any other AAE that would represent a combination of internally or externally mixed BC and BrC

Printer-friendly version

Discussion paper

as long as the total absorption behavior dose not significantly differ from a power (or exponential) behavior in the selected wavelength range.

Page 9, section 3.1 and Figure 5. A few questions / comments.

1) The relative errors are large in extinction due to absorption (and in imaginary refractive index) and small in extinction due to scattering and in total extinction (and real refractive index). One would guess that large errors in absorption would translate directly into large errors in SSA, which is directly proportional to absorption, yet errors in SSA are much smaller. Some explanation is warranted. The SSA is directly related to extinction and also to either the scattering or the absorption in a straightforward way that depends on how you choose to calculate it. The errors on SSA are also related in the same way. The closer the SSA is to a value of 1 the more its errors are related to the errors in the scattering. The closer the SSA is to a value of 0 the more its errors are related to the errors in the absorption. An example for the following “true” values: Ext = 100, Abs = 5 (Sca = 95, SSA = 0.95) and with errors of +-5% on Ext and +-500% on Abs. The errors on the Sca and on the SSA would be about +-25%. for the following “true” values: Ext = 100, Abs = 50 (Sca = 50, SSA = 0.5) and with errors of +-5% on Ext and +-500% on Abs. The errors on the Sca and on the SSA would follow the errors on the Abs, namely about +-500%.

2) Relative errors are given with no indication of the sign of the errors. Does this analysis reveal any systematic deviation, or are the errors simply distributed about zero?

The distribution of errors around zero is strongly related to the instrumental wavelengths used, to the wavelength range chosen for the extrapolation and to the wavelength range presented (300-400 nm, 400-500 nm etc.). In our analysis errors in some presented wavelength ranges were skewed to negative error and some to positive. Because of the subjective nature of the error distribution we chose to preset its absolute value and not its direction.

Printer-friendly version

Discussion paper

Interactive
comment

3) If I understand this procedure correctly, the measured values are assigned to the correct value in the synthetic data for the purpose of testing the retrieval. Please correct if I have misunderstood. What would be the effect and / or additional error in the retrieval if measurement uncertainty were considered (i.e., measurements assigned to values different from the synthetic data according to the measurement uncertainty distribution)?

The errors in the retrieval already include the random errors; these were applied to the synthetic data points before the retrieval procedure. In section 2.4.1 the sentence: "An additional error was assigned randomly out of a normal distribution with  $\pm 2\%$  (1 standard deviation)" was changed to: "An additional error was assigned to each calculated  $\alpha_{\text{ext}}$ ,  $\alpha_{\text{sc}}$  and  $\alpha_{\text{abs}}$  at the instrumental wavelengths, randomly out of a normal distribution with  $\pm 2\%$  (1 standard deviation)".

Page 10, line 6-7: Wouldn't one expect agreement between the measurement and retrievals at the single wavelength to which the aerosol absorption is constrained in the retrieval? Suggest adding the phrase "as expected" or equivalent to indicate this.

The sentence was revised to: "The good agreement between measured and retrieved SSA is an indication that the broadband extinction retrieval procedure has little to no error at the wavelength at which the aerosol absorption is constrained."

Moosmuller, H., Chakrabarty, R. K., Ehlers, K. M., and Arnott, W. P.: Absorption angstrom coefficient, brown carbon, and aerosols: Basic concepts, bulk matter, and spherical particles, *Atmos. Chem. Phys.*, 11, 1217-1225, 2011.

Please also note the supplement to this comment:

<http://www.atmos-meas-tech-discuss.net/amt-2016-66/amt-2016-66-AC5-supplement.pdf>